# CausalPlan: Empowering Efficient LLM Multi-Agent Collaboration Through Causality-Driven Planning

## Abstract

Large language model (LLM) agents often generate causally invalid plans in collaborative tasks due to their reliance on surface-level correlations rather than grounded causal reasoning. This limitation undermines their performance in terms of coordination and planning in dynamic environments. We address this challenge with CausalPlan, a framework that integrates explicit structural causal reasoning into the LLM planning process. At the core of CausalPlan is the Structural Causal Action (SCA) model, which learns a causal graph from agent trajectories to capture how prior actions and current environment states influence future decisions. This model is then used to inform the planning process, shaping proposed LLM-generated plans through causal scoring, reweighting, and fallback to grounded alternatives when needed. By embedding this causal knowledge directly into the decision loop, CausalPlan constrains planning to intervention-consistent behaviors without requiring fine-tuning. We evaluated CausalPlan on the Overcooked-AI benchmark across five multi-agent coordination tasks and four LLMs of varying sizes: Gemma-7B, Llama-8B, Qwen-14B and Llama-70B. Experimental results show that CausalPlan consistently reduces invalid actions and improves collaboration in both AI-AI and human-AI settings, outperforming strong reinforcement learning baselines. Our findings highlight the value of causality-driven planning for deploying efficient, interpretable, and generalisable multi-agent LLM systems.

## 1 Introduction

Large Language Models (LLMs) have demonstrated significant success across various natural language processing tasks (Achiam et al., 2023; Zhao et al., 2023b; Guo et al., 2025). Recently, there has been growing research interest in using LLMs as decision makers, particularly within multi-agent frameworks for executing interactive planning tasks, with notable works including integrated pipelines for cooperative tasks (Zhang et al., 2023a), graph-based coordination (Qian et al., 2024), and human-AI collaboration frameworks (Zhang et al., 2024a).

A major challenge in multi-agent learning is zero-shot multi-agent coordination, developing generalized agents capable of collaborating with a wide range of previously unseen partners, including humans (Legg & Hutter, 2007; Hu et al., 2020). LLM-based agents, trained on vast and diverse datasets that contain rich common-sense knowledge, have emerged as a promising solution to this challenge. Compared to traditional multi-agent reinforcement learning (RL) methods—which often struggle with generalization and sample inefficiency—LLMs demonstrate impressive performance in collaborative tasks (Zhang et al., 2024a). However, despite these strengths, a persistent limitation remains: LLM agents often lack causal reasoning ability (Joshi et al., 2024; Chi et al., 2024). This shortcoming leads them to select causally invalid actions that violate *causally physical constraints*, actions that are absent or cannot be executed under the given task constraints, and ignoring *temporal dependencies*, producing sequences of actions that do not respect the natural order of cause and effect. This problem is particularly pronounced in smaller open-source LLMs due to their limited capacity and narrower training coverage. As shown in Fig. 1(a), our evaluation of multiple open-source LLMs with varying parameter sizes demonstrates that even Llama-70B produces a substantial number of invalid actions. Despite this limitation, such models remain highly attractive for enterprise and resource-constrained settings because of their accessibility, controllability, and lower deploy-

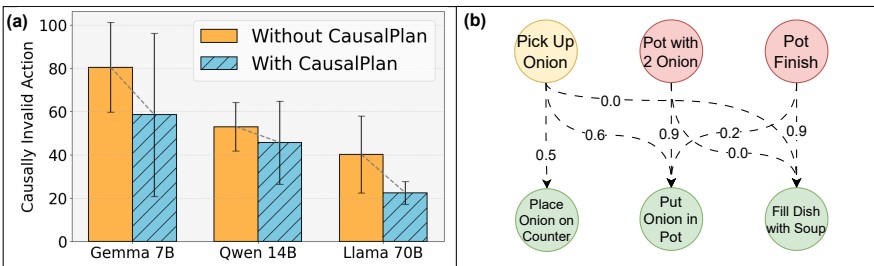

Figure 1: **(a)** Evaluation on the Overcooked Cramped Room layout showing how the number of causally invalid actions changes with LLM size, averaged over four seeds. CausalPlan significantly reduces the number of invalid moves. **(b)** Simplified causal graph discovered by CausalPlan for the same layout. Yellow and red nodes indicate parent actions and states, respectively, while green nodes denote child actions. "Pick Up Onion" strongly influences "Put Onion in Pot" (0.6) and "Place Onion on Counter" (0.5), but not "Fill Dish with Soup" (0). The state "Pot with 2 Onions" strongly drives "Put Onion in Pot" (0.9), while "Pot Finished" strongly influences "Fill Dish with Soup".

ment costs. However, their higher incidence of causally invalid actions can significantly undermine performance. Although previous work has tried to improve LLM planning with causal knowledge, it primarily focuses on single-agent settings and relies on LLMs to infer causal relationships from observations or provide the causal graph as part of the planning prompt (Yu & Lu, 2025; Chen et al., 2025). These approaches are limited because they depend on the robustness of the LLM's causal reasoning and inference ability, which can vary significantly between models and prompts. This motivates the need to integrate causal knowledge directly into the decoding process, rather than relying on prompt engineering, so that LLM action planning is grounded in cause–and–effect structure and yields more reliable coordination in multi-agent settings. Ultimately, our aim is to answer the question of: *"How can we systematically align LLM action planning with explicit causal knowledge to ensure reliable and effective collaboration in multi-agent settings?"*

To answer the question, we introduce the CausalPlan framework, grounded in the study of causality (Pearl, 2009). In causality, causal relationships can be represented by a causal graph $\mathcal{G}$, with the structural causal model (SCM) a formal framework that defines how each variable is generated from its parent variables in the graph (see Fig. 1 (b) for an example) (Pearl, 2009). An SCM can be identified through causal discovery, and once identified, an SCM supports causal inference for downstream tasks (Pearl, 2009). CausalPlan translates these principles of causality into the multi-agent LLM planning setting. The framework consists of two key phases inspired by the discovery and inference processes: *Causal Action Structure Learning* and *Agent Planning with Causal Knowledge*. In Causal Action Structure Learning, we introduce a Structural Causal Action (SCA) model, an extension of SCM tailored to capture the causal relationships between previous actions of agents, current states of both agents, and future actions. Importantly, the SCA model characterizes causal influence at the policy-level within the agent's decision process, rather than causal dynamics at the environment-level in the Pearl sense (Pearl, 2009). Its purpose is not to model the true causal mechanisms of the environment, but to extract a stable and interpretable dependency structure that can guide and refine the LLM's action selection. For example, before serving a plate of soup (future action), one must first fill the dish with soup (past action); similarly, if the partner agent is already carrying a filled dish (partner state), the controlled agent should focus on complementary actions rather than duplicating effort. Once discovered, the SCA produces a *Causal Action Matrix $\mathcal{M}$*, which encodes causal relationships as causal scores and can be queried during planning using the current state and past actions of the agents.

In the Agent Planning with Causal Knowledge phase, we align the LLM decoding process with the scores in $\mathcal{M}$ to prevent causally invalid actions. To achieve this, we introduce two complementary strategies: *Causal-Aware Planning* and *Causal Backup Plan*. The Causal-Aware Planning module adjusts the LLM's action probabilities by reweighting them with causal scores and then resampling to select actions that follow the natural order of cause and effect. When all candidate actions proposed by the LLM violate the causally physical constraints of the task, the *Causal Backup Plan* module adjusts by selecting the action with the highest causal probability as the next action.

We evaluate CausalPlan on the Overcooked-AI benchmark (Carroll et al., 2019), a standard testing suite for multi-agent, using four open-source LLMs—Gemma-7B, Llama-8B, Qwen-14B, and Llama-70B—across both AI-AI and human-AI collaboration settings. Empirical results show that CausalPlan consistently improves planning performance and reduces invalid actions, even for the smallest LLMs without fine-tuning. Our main contributions are: (i) We identify a core failure mode of LLM agents in multi-agent collaboration generating causally invalid actions and propose causally aligned planning as a principled remedy; (ii) We introduce CausalPlan, a two-phase framework that integrates causal discovery and inference to enhance open-source LLM agent planning and collaboration; (iii) We demonstrate, through extensive experiments, that CausalPlan improves performance across multiple model sizes and collaboration scenarios, outperforming strong RL baselines.

## 2 Preliminaries

**Markov Decision Process.** A two-player Markov Decision Process (MDP) is defined as $(\mathcal{S}, \{\mathcal{A}^i\}, P, \gamma, R)$, where $\mathcal{S}$ is the state space, $\mathcal{A}^i$ is the action set for agent $i \in \{1, 2\}$, $P$ defines the transition dynamics, $\gamma \in [0, 1)$ is the discount factor, and $R : \mathcal{S} \times \mathcal{A} \mapsto \mathbb{R}$ is the reward function where $\mathcal{A} = \mathcal{A}_1 \times \mathcal{A}_2$ is the joint action space. We assume a factored state space $\mathcal{S} = \mathcal{S}^{\text{agent}} \times \mathcal{S}^{\text{env}}$, where $\mathcal{S}^{\text{agent}}$ is the state of the agent (both agent 1 and 2) and $\mathcal{S}^{\text{env}}$ the state of the environment. Let $S = |\mathcal{S}|$ and $A = |\mathcal{A}|$ denote the dimensions of $\mathcal{S}$ and $\mathcal{A}$, respectively. At each timestep $t$, each agent $i \in \{1, 2\}$ observes the current state $s_t = (s_t^{\text{agent}}, s_t^{\text{env}})$ and selects an action according to its policy $\pi^i(a_t^i \mid s_t)$, forming the joint action $a_t = (a_t^1, a_t^2)$. A trajectory is given by $\tau = (s_1, a_1, s_2, a_2, \dots)$, and the objective is to maximize the cumulative expected reward $\mathbb{E}\left[\sum_t R(s_t, a_t)\right]$. In our two-agent setting, one of the agents is the controlled agent (an LLM-based agent), while the other serves as its partner.

**Causality and Structural Causal Model.** Causality studies the relationships between variables and events (Pearl, 2009). The SCM framework represents causal relationships in a system, where for a set of variables $V = \{V_1, \dots, V_M\}$, each variable $V_i$ is defined as $V_i := f_i(\text{Pa}_{\mathcal{G}}(V_i), \varepsilon_i)$, with $\{f_1, f_2, \dots, f_M\}$ being generating functions, $\text{Pa}_{\mathcal{G}}(V_i)$ the parents of $V_i$ in the causal graph $\mathcal{G}$, and $\{\varepsilon_1, \dots, \varepsilon_M\}$ noise terms (Pearl, 2009). The directed acyclic graph (DAG) causal $\mathcal{G} = \{V, E\}$ contains edges $e_{ji} \in E$, where $e_{ji} = 1$ indicates that $V_j$ causes $V_i$, and $e_{ji} = 0$ otherwise (Pearl, 2009). SCMs are often learned from data by modeling the generating functions $f_i$ as neural networks parameterized by generating parameters $\delta$ (Ke et al., 2019; Peng et al., 2022; Zhang et al., 2023b), with causal edges $e_{ji} = 1$ if the binary adjacency indicator $\eta_{ji}$ is higher than a confidence threshold (Ke et al., 2019; Peng et al., 2022; Zhang et al., 2023b).

## 3 Method

Our CausalPlan is a two-phase framework (Fig. 2). In Phase 1, *Causal Action Structure Learning*, we construct the SCA model and derive from it *Causal Action Matrix $\mathcal{M}$*. In Phase 2, *Agent Planning with Causal Knowledge*, we align the LLM's planning process with the causal scores in $\mathcal{M}$, using them to guide the action selection process. At each planning step $t$, we first provide the current observation $s_t$ to the LLM agent and prompt it to analyze the observation. Both the observation $s_t$ and the analysis are then used as inputs for a second prompt, where the agent is asked to generate a set of candidate actions (details of the prompt are in Appx. C.2.1). We, then, leverage $\mathcal{M}$ to modify the agent's plan selection, either through the Causal-Aware Planning module or the Causal Backup Plan module (see Appx. C for the full algorithms).

### 3.1 Causal Action Structure Learning

The goal of the first phase is to construct an SCA model, capturing the causal graph $\mathcal{G}$, where the previous action $a_{t-1}$ and the current state $s_t$ are the parent nodes, and the next action $a_t$ is the child node. Unlike prior work, which typically focuses on modeling state transitions or rewards (Zhang et al., 2023b; 2024b), our approach explicitly treats the action as a child node. This novel formulation allows the agent to reason causally about how past actions and current states influence future actions, providing a new perspective on decision-making dynamics.

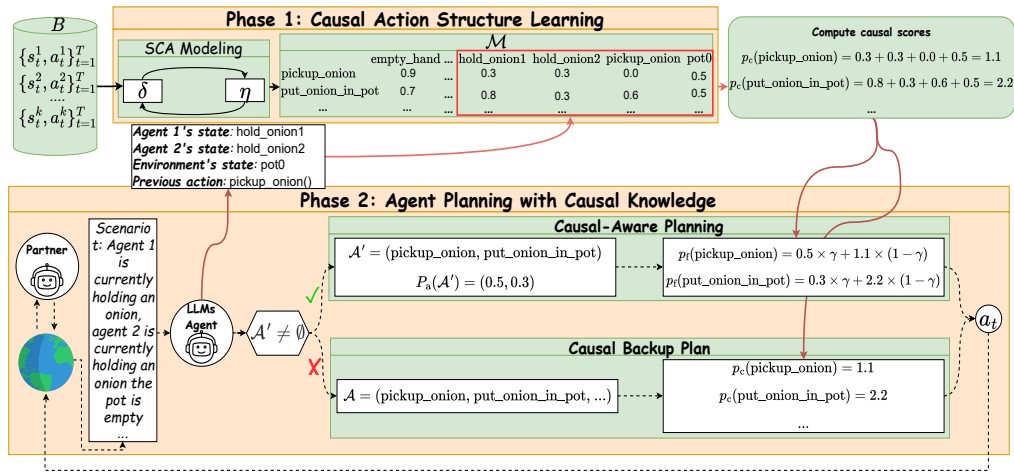

Figure 2: **Overview of the CausalPlan Framework**. The process begins with a dataset $B$ collected by a behavior policy $\pi_\beta$. In Phase 1 (*Causal Action Structure Learning*), we train the Structural Causal Action (SCA) Model by optimizing generating ($\delta$) and structural ($\eta$) parameters, yielding the *Causal Action Matrix* $\mathcal{M}$, which encodes causal influence from states and past actions to future actions. In Phase 2 (*Agent Planning with Causal Knowledge*), an LLM receives scenario $t$ and proposes candidate actions $\mathcal{A}'$. If $\mathcal{A}' \neq \emptyset$, *Causal-Aware Planning* adjusts LLM probabilities; if $\mathcal{A}' = \emptyset$, *Causal Backup Plan* selects the most probable past action via $\mathcal{M}$. Black solid arrows denote causal training; dashed arrows denote LLM inference, and red arrows denote causal knowledge consultation. The red box represents the causal score extraction for each potential next action, where the score is computed as the sum of causal contributions from the current state and previous action.

**Data Preparation.** To facilitate the process of SCA modeling, we collect a dataset $B = \left\{ \{ (s_t^k, a_t^k) \}_{t=1}^T \right\}_{k=1}^N$ containing actions that have been executed successfully in the environment, using a behavior policy $\pi_\beta$. We, then, factorize and discretely encode the states and actions, which are collected in text form into a binary-encoded representation suitable for causal analysis: $s_t = [s_{t,1}, \ldots, s_{t,S}] \in \{0,1\}^S$, $a_t = [a_{t,1}, \ldots, a_{t,A}] \in \{0,1\}^A$, where each component $s_{t,j}$ and $a_{t,i}$ is a binary indicator representing whether a particular state feature or action is active (1) or inactive (0) (refer to Appx. C.1 for details). The assumption of factorized states and actions is a common assumption in most causal RL research (Ke et al., 2019; Yu & Lu, 2025).

**Causal Modeling.** The SCA model can be represented as:

$$a_i = f_i \left( \mathrm{Pa}_{\mathcal{G}}(a_i), \varepsilon_{a_i} \right) \tag{1}$$

for $i \in \{1, 2, \ldots, A\}$, where $\mathrm{Pa}_{\mathcal{G}}(a_i)$ denotes the parent nodes for $a_i$ in the causal graph $\mathcal{G}$. The function $f_i$ is a neural network parameterized by the generating parameter $\delta$, while the causal relationships of each graph are governed by the structural parameters encoded by binary adjacency indicators $\eta_{ji}$. The loss function to optimize these parameters is: $L(\delta, \eta) = L_{\mathrm{causal}}(\delta, \eta) + L_{\mathrm{reg}}(\eta)$, where:

$$L_{\mathrm{causal}} = \mathbb{E}_{(a_{t-1}, s_t, a_t) \sim B} \left[ -\sum_{i=1}^A \log P \left( a_{t,i} \mid s_t, a_{t-1}; \delta, \eta \right) \right]. \tag{2}$$

$L_{\mathrm{reg}}(\eta)$ is a negative-log-prior penalty imposed on the adjacency indicators to discourage spurious edges and avoid overfitting to unlikely causal links. Let $P(e_{ji} = 1)$ be the prior probability for any

edge. Then

$$L_{\text{reg}} = -\lambda \sum_{i,j} \eta_{ji} \log P(e_{ji} = 1) \,, \tag{3}$$

where $\lambda > 0$ controls the relative contribution of each penalty term. Including an edge $\eta_{ji} = 1$ incurs a cost $-\log P(e_{ji} = 1)$, so only edges with high prior belief are preferred.

In causal inference, identifiability, referring to the ability to recover causal effects from data (Pearl, 2009) uniquely, is crucial for valid causal inferences. In our setting, identifiability intuitively ensures that the underlying causal structure and decision policy could, in principle, be recovered from observed trajectories. We emphasize that the following proposition and its proof serve as a conceptual illustration for the SCA model used in practice:

**Proposition 1** (Identifiability (Conceptual Illustration)). *Suppose that the state $s_t$ and previous action $a_{t-1}$ are observable, while the next action $a_t$ is observable during training and unobserable during inference, and they form a Markov Decision Process (MDP) as described in Eq. 1. Then, under the global Markov condition and the faithfulness assumption given a large enough dataset $B$, the next action $a_t$ is identifiable, as well as the causal structure characterized by the binary masks $\eta$ and the transition dynamics $f$.*

*Proof.* See Appx. A.

$\square$

**Causal Action Matrix construction.** We then construct the matrix $\mathcal{M} \in \mathbb{R}^{A \times (S+A)}$ that encodes the causal score of selecting each action given the current state and past actions. Each row of the matrix corresponds to a possible next action, and each column corresponds to a state or past action feature. Each entry $(i, j)$ of the matrix represents the probability that there is causal influence from state or action feature $j$ to action $i$, given by the learned structure parameter $\eta_{ji}$.

A query $\mathcal{M}(s_t, a_{t-1}, a)$ returns the causal score $p_c(a) = \sum_{j \in J} \eta_{ji}$ where $J = \text{Active}(s_t, a_{t-1}) \subseteq \{1, \ldots, S + A\}$ denote the set of column indices corresponding to the features that are "active" in the current state $s_t$ and the previous action $a_{t-1}$, and $i$ is the row index corresponding to action $a$ (details refer to Appx. C.2.2). To enforce a partial DAG structure in the causal graph, we compare the coefficients for each pair of mutually connected nodes in $\mathcal{M}$ and set the smaller coefficient to zero, which removes 2-cycles (Pearl, 2009).

## 3.2 AGENT PLANNING WITH CAUSAL KNOWLEDGE

At each planning step, instead of directly generating the next action $a_t$ given the historical trajectory $h_t = (s_1, a_1, s_2, a_2, \ldots, a_{t-1}, s_t)$, we require the LLM-based agent to consider alternative scenarios and select the action that aligns with the causal scores in the matrix $\mathcal{M}$. Firstly, we sample from the LLM a set of candidate actions $\mathcal{A}' = \left\{ a_1', a_2', \ldots, a_{|\mathcal{A}'|}' \right\} \subseteq \mathcal{A}$. Each of these actions will come with a probability of being sampled by the LLM, which we denote as $p_a(a_m')$. Next, we verify whether the sampled actions, assuming access to a feasibility verifier for candidate actions, comply with the environment's instructions and physical constraints. If the set $\mathcal{A}' \neq \emptyset$ (there are valid candidates), we follow the Causal-Aware Planning module to find the most suitable action that follows causal temporal dependencies; otherwise, we use the Causal Backup Plan for the causal backup mechanism.

### 3.2.1 CAUSAL-AWARE PLANNING

Given the set $\mathcal{A}'$ with their associated probabilities $P_a(\mathcal{A}')$, we aim to integrate the causal scores from the model $\mathcal{M}$. We extract the causal score for each action $p_c(a') = \mathcal{M}(s_t, a_{t-1}, a'), \forall a' \in \mathcal{A}'$, to form the set $P_c(\mathcal{A}') = \left\{ p_c(a_1'), p_c(a_2'), \ldots, p_c(a_{|\mathcal{A}'|}') \right\}$ (details in Appx. C.2.2). The updated individual action probabilities are computed as the weighted sum of the LLM sampling probability and the causal score:

$$p_f(a_m') = \gamma \cdot p_a(a_m') + (1 - \gamma) \cdot p_c(a_m'), \tag{4}$$

where $\gamma$ is the weight hyperparameter. We apply the softmax function to all values of $p_\text{f}(a'_m)$ to normalize the probabilities, which allows us to get the final probability set:

$$P_\text{f}(\mathcal{A}') = \left\{ p_\text{f}(a'_1), p_\text{f}(a'_2), \ldots, p_\text{f}(a'_{|\mathcal{A}'|}) \right\}, \quad \sum_{k=1}^{|\mathcal{A}'|} p_\text{f}(a'_m) = 1 \tag{5}$$

The sampled action set $\mathcal{A}'$ may contain redundant actions, so we apply a method to identify and merge these duplicates by summing their probabilities (details in Appx. C.2.4). This yields a reduced set $\mathcal{A}'^*$ with updated probabilities $P_\text{f}^*$, from which we sample the next action:

$$a_t \sim \text{Categorical}\left( \left[ p_\text{f}^*(a'_1), p_\text{f}^*(a'_2), \ldots, p_\text{f}^*(a'_{|\mathcal{A}'^*|}) \right] \right). \tag{6}$$

### 3.2.2 CAUSAL BACKUP PLAN

In the second case, when all candidates are invalid $\mathcal{A}' = \emptyset$, existing methods often apply a fall-back strategy by prompting the agent to re-plan (Zhang et al., 2024a). However, such strategies may fail when the agent persistently hallucinates, for instance, when the state stays unchanged. Inspired by human behavior under uncertainty, choosing the action that we are most familiar with, we propose a recovery mechanism that leverages past causality knowledge. Instead of immediately re-planning, we ask the agent to retrieve the causal score for all actions $a \in \mathcal{A}$ by querying $p_\text{c}(a) = \mathcal{M}(s_t, a_{t-1}, a), \forall a \in \mathcal{A}$, (details in Appx. C.2.2). This yields a probability distribution: $P_\text{c}(\mathcal{A}) = \{ p_\text{c}(a_1), p_\text{c}(a_2), \ldots, p_\text{c}(a_A) \}$. We then greedily select the next action given by:

$$a_t = \underset{a \in \mathcal{A}}{\arg\max}\, P_\text{c}(a), \tag{7}$$

i.e., the action deemed most reliable according to past causal knowledge. Only if this action fails do we then ask the agent to re-plan.

## 4 EXPERIMENTS

### 4.1 EXPERIMENTAL SETUP

We use the Overcooked-AI environment suite (Carroll et al., 2019) as our main testing platform. This suite comprises five distinct layouts: *Cramped Room* (CR), *Asymmetric Advantages* (AA), *Coordination Ring* (COR), *Forced Coordination* (FC), and *Counter Circuit* (CC) (details of the environments in Appx. D.2). Each layout evaluates distinct aspects of multi-agent coordination, making this environment a standard for evaluating agent collaboration. Our experiments aim to demonstrate that CausalPlan can improve planning for various open-source LLMs and, thus, better collaboration. Specifically, we use `gemma-1.1-7b-it` (Gemma-7B), `Meta-Llama-3-8B-Instruct` (Llama-8B), `Qwen2.5-14B-Instruct-1M` (Qwen-14B), and `Llama-3.3-70B-Instruct` (Llama-70B). These open-source models are integrated into ProAgent (Zhang et al., 2024a), a framework that leverages advanced prompting techniques (ReAct (Yao et al., 2023) and Reflexion (Shinn et al., 2023)), upon which we apply CausalPlan to refine the planned actions. Additionally, we use `Cohere/command-r` Cohere (2024), a 35-billion-parameter model, to generate the analysis of the observation in our two-prompt input (refer to Appx. C.2.1 for details).

In Sect. 4.2, we compare the performance of LLM agents with their performance when enhanced with CausalPlan. Our agent is evaluated alongside baseline partner AI agents (see next paragraph). In these experiments, our agents play as Player 1 and the baseline agents as Player 0. An effective agent should demonstrate strong performance in collaboration with all other partners. We also compare our CausalPlan agent with the Llama-70B backbone against the baseline agents playing as both Player 0 and Player 1. In Sect. 4.3, we evaluate the performance of CausalPlan agents when collaborating with human-like agents (collected using Behavior Cloning (Li et al., 2023b)). In Sect. 4.4, we evaluate different components of CausalPlan and in Sect. 4.6 we analyze the benefits of integrating causal knowledge. In Sect. 4.4, we provide experimental comparisons with a natural non-causal supervised baseline. In the Appendix, we provide additional experiments such as parameter tuning $\gamma$ (Appx. D.6), different data collection policies $\pi_\beta$ (Appx. D.7), time complexity

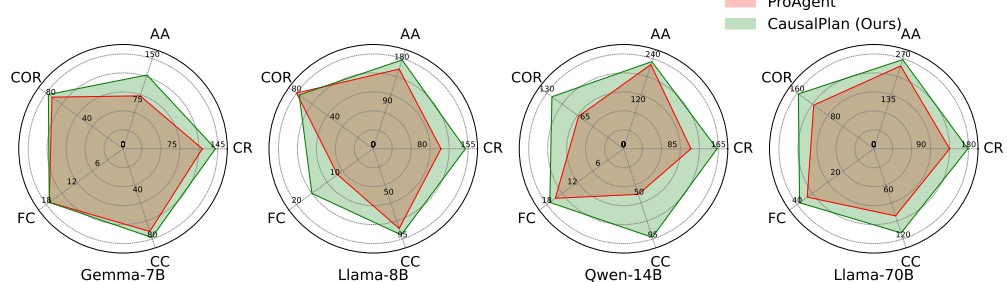

Figure 3: Performance of different backbones with and without CausalPlan across various layouts. In these experiments, we use the LLM agent as Player 1, allowing it to collaborate with all other baselines (described in Sect. 4.1) for 400 timesteps and report the average of three different seeds.

analysis (Appx. D.10), the causal matrix $\mathcal{M}$ (Appx. D.9), and a verification of the causal graph found (Appx. E.1).

**Baselines.** The baselines include traditional RL methods designed for zero-shot human and AI coordination. These baselines have achieved notable results in the field, including SP (Tesauro, 1994; Carroll et al., 2019), PBT (Jaderberg et al., 2017), FCP (Strouse et al., 2021), MEP (Zhao et al., 2023a), COLE (Li et al., 2023b) (refer to Appx. D.3 for baseline details).

We also evaluate CausalPlan in the Crafter environment (Hafner, 2021), a long-horizon planning benchmark, where it outperforms Causal-Aware LLMs (Chen et al., 2025) (the state-of-the-art causal prompting approach). Due to space constraints, detailed results are deferred to Appx. E.

### 4.2 AI PARTNER EVALUATION

**Enhancing open-source LLM performance using CausalPlan** We evaluate whether CausalPlan improves open-source LLM performance in collaboration tasks, as shown in Fig. 3 and detailed in Appx. D.4 Tab. 4. CausalPlan improves performance models, with significant gains seen in Qwen-14B (29.04%) and Llama-70B (22.42%). In terms of layouts, the most substantial improvements were found in the settings CR (20.83%) and COR (19.13%). Furthermore, CausalPlan also provided notable benefits for larger LLMs, such as Llama-70B, demonstrating its potential to enhance performance even at scale.

**Comparison with state-of-the-art RL baselines.** We evaluate the performance of our top-performing agent (Llama-70B backbone) against the set of SOTA baseline RL agents. The results, presented in Tab. 1, show that our agent consistently ranks among the top performers across different layouts (highest score in three out of five layouts and second in one additional layout). The most significant performance gaps between our method and the next best baseline are observed in the AA layout, showing a 63% advantage. We attribute the underperformance in CR to the simplicity of the task, which does not require causal knowledge. These results demonstrate that, when equipped with CausalPlan, open-source LLM agents can outperform state-of-the-art RL agents in various tasks, highlighting the effectiveness of integrating causal reasoning into cooperative LLM-based agents.

### 4.3 HUMAN PARTNER EVALUATION

To evaluate human collaboration, we performed an experiment using human proxy partners, with the results shown in Fig. 4. In this experiment, our CausalPlan framework utilizes Llama-70B as the backbone LLM. As shown, our agent (green bars) outperforms all baselines in 8 out of 10 configurations. On average across all layouts, it achieves approximately a 30% improvement over ProAgent (red bars), and outperforms the strongest RL baseline (COLE) by approximately 32%. To further validate these improvements, we conducted statistical analyses using paired $t$-tests and corresponding $p$-values. The results (Appx. D.5 Tab. 6) show that CausalPlan consistently achieves higher $t$ values than ProAgent when compared against the best RL method. Direct comparison in Tab. 7) reveals statistical significance ($p < 0.05$) in 30% of the cases (CR-P0, AA-P1, COR-

Table 1: Average performance (mean ± std) of baseline agents and CausalPlan (Ours) across layouts using Llama-70B. Results are averaged over both player positions and three seeds (400 timesteps each). Best and second-best results are in **bold** and underlined, respectively. Detailed performance of playing as Player 0 or Player 1 is provided in Appx. Tab. 5.

| Layout | Baseline AI Agents | | | | | CausalPlan (Ours) |
|---|---|---|---|---|---|---|
| | **SP** | **PBT** | **FCP** | **MEP** | **COLE** | |
| **CR** | $162.0 \pm 10.0$ | $168.0 \pm 5.0$ | **$194.0 \pm 10.1$** | $178.0 \pm 16.1$ | $153.4 \pm 12.5$ | $172.7 \pm 4.2$ |
| **AA** | $184.0 \pm 17.5$ | $168.0 \pm 15.4$ | $176.6 \pm 15.0$ | $167.3 \pm 5.8$ | $185.3 \pm 15.1$ | **$258.7 \pm 16.4$** |
| **CC** | $56.7 \pm 9.2$ | $52.0 \pm 14.0$ | $63.4 \pm 10.5$ | $50.0 \pm 16.1$ | $90.6 \pm 10.1$ | **$112.6 \pm 7.6$** |
| **COR** | $120.7 \pm 11.0$ | $139.4 \pm 10.1$ | $130.7 \pm 6.2$ | **$160.7 \pm 7.2$** | $153.4 \pm 4.6$ | $156.6 \pm 3.2$ |
| **FC** | $18.0 \pm 4.6$ | $40.6 \pm 10.3$ | $42.0 \pm 7.2$ | $30.4 \pm 5.4$ | $44.6 \pm 7.0$ | **$53.9 \pm 14.9$** |

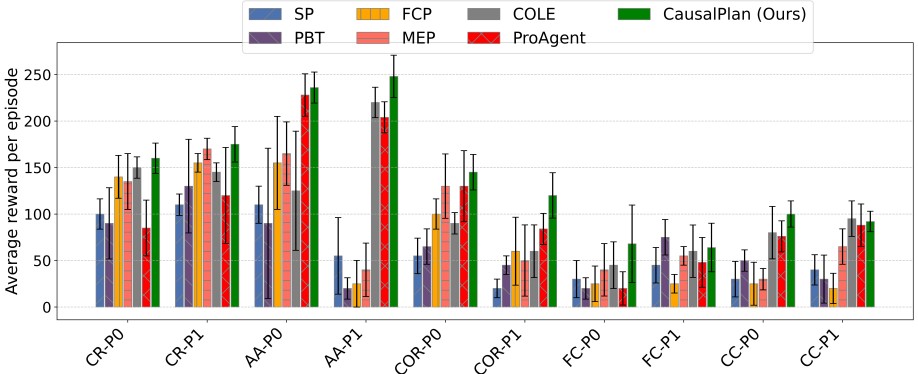

Figure 4: Experiments with a human proxy partner. Results show the mean and variance averaged over using five different BC policies as the partner (each running for 400 timesteps). "P0" denotes the controlled AI agent acting as Player 0, and vice versa.

P1), with another 30% (CR-P1, FC-P0, CC-P0) showing marginal significance ($0.05 < p < 0.2$). Importantly, performance never degrades when CausalPlan is included. These findings confirm that the observed improvements are statistically reliable.

### 4.4 IMPACT OF CAUSALPLAN COMPONENTS

In this section, we investigate the individual contributions of each component within the CausalPlan framework. First, we compare the use of a single prompt (Zhang et al., 2024a), for both observation analysis and planning, against our two-prompt setup, where one prompt is dedicated to analysis and the other to planning. This comparison helps isolate whether performance gains come from the embedded causal knowledge. As shown in Tab. 2, the performance between the single-prompt and two-prompt configurations is nearly identical, with only a slight improvement when using our two-prompt. Second, we examine the effect of the Causal Backup Plan module. CausalPlan without the backup action still outperforms the two-prompt variant by 27%, but falls short of the full framework by 7%. This highlights the significance of the backup mechanism to avoid scenarios in which the agent fails to select actions as instructed.

### 4.5 COMPARISON TO NON-CAUSAL SUPERVISED CONDITIONAL MODELS

To demonstrate that CausalPlan provides stronger guidance than a non-causal conditional model, we evaluate a supervised baseline trained to estimate $P(a_t \mid s_t, a_{t-1})$ from MEP data. As shown in Tab. 3, combining this learned model with the backbone Llama-70B (which we refer to as $\text{MEP}_{\text{guided}}$) consistently degrades performance (16.4% drop on AA-P1 and 10.1% drop on CC-P1), indicating that a non-causal conditional model constrains the LLM to suboptimal demonstrated actions. Using

Table 2: Ablation studies were conducted on the CR layout using Llama-8B. "1-Prompt" uses a single prompt for observation and planning, as in ProAgent; "2-Prompt" uses our modified dual-prompt method. "CausalPlan (no CBP)" omits the Causal Backup Plan component.

| Methods | Baseline AI Agents | | | | | Average Results |
|---|---|---|---|---|---|---|
| | SP | PBT | FCP | MEP | COLE | |
| **1-Prompt (ProAgent)** | $86.7 \pm 41.6$ | $66.7 \pm 63.4$ | $180.0 \pm 20.0$ | $106.7 \pm 75.7$ | $113.3 \pm 11.5$ | $110.7 \pm 12.8$ |
| **2-Prompt** | $73.3 \pm 30.5$ | $93.3 \pm 57.7$ | $180.0 \pm 0.0$ | $126.7 \pm 11.5$ | $126.7 \pm 23.1$ | $121.3 \pm 2.3$ |
| **CausalPlan (no CBP)** | $113.3 \pm 23.1$ | $146.7 \pm 46.2$ | $160.0 \pm 34.6$ | $133.3 \pm 11.5$ | $153.3 \pm 23.1$ | $141.3 \pm 12.9$ |
| **CausalPlan (Full)** | $126.7 \pm 30.6$ | $133.3 \pm 30.5$ | $160.0 \pm 40.0$ | $166.7 \pm 41.6$ | $166.7 \pm 23.1$ | $\mathbf{150.7 \pm 2.3}$ |

Table 3: Performance comparison between the backbone Llama-70B policy, the non-causal supervised baselines ($\text{MEP}_{\text{guided}}$ and $\text{MEP}_{\text{backup}}$), and our method CausalPlan. The $\text{MEP}_{\text{guided}}$ baseline combines the learned conditional model $P(a_t \mid s_t, a_{t-1})$ with the backbone by averaging action probabilities. The $\text{MEP}_{\text{backup}}$ baseline replaces CausalPlan's backup mechanism with the conditional model.

| Layout | Llama-70B | $\text{MEP}_{\text{guided}}$ | $\text{MEP}_{\text{backup}}$ | CausalPlan |
|---|---|---|---|---|
| AA-P1 | $248.0 \pm 22.7$ | $207.3 \pm 19.4$ | $257.3 \pm 9.2$ | $\mathbf{266.7 \pm 16.7}$ |
| CC-P1 | $89.3 \pm 32.3$ | $80.3 \pm 9.2$ | $90.7 \pm 12.9$ | $\mathbf{112.0 \pm 6.9}$ |

the conditional model only as a backup ($\text{MEP}_{\text{backup}}$) yields small gains over the backbone LLM but remains notably worse than CausalPlan. We attribute this gap to a fundamental difference in what each method can correct. $\text{MEP}_{\text{backup}}$ can only correct physically invalid actions, whereas CausalPlan additionally enforces the correct temporal dependencies between actions. Ensuring that the agent selects actions in the right causal order is equally crucial—especially in coordination tasks where timing and sequence are important. Overall, these results show that non-causal supervised models cannot surpass the behavior policy, while CausalPlan provides a causal structure that enables consistent improvements across layouts.

### 4.6 BENEFITS OF CAUSAL INTEGRATION

We analyze the behavior of Llama-8B, with and without CausalPlan, in the CR layout, where our method achieves a substantial $+36.1\%$ improvement (see Appx. D.8 for detailed analysis). This analysis highlights two key benefits of causal integration.

**(1) Physically invalid actions.** Without causal guidance, the agent frequently makes invalid calls to pick up an object while already holding an object. CausalPlan reduces these physically invalid actions by 18%, while simultaneously increasing valid calls made with an empty hand by 17%. This demonstrates that CausalPlan not only suppresses impossible actions but also systematically promotes temporally valid ones.

**(2) Poor coordination.** Coordination failures are further mitigated. When the pot is nearly full and the partner agent already has an onion, the baseline still selects redundant actions to pick up onion. With CausalPlan, these cases drop to zero, indicating that the agent learns to anticipate teammate states and avoid conflicting behaviors. This complete elimination of redundant pickups reflects a higher level of situational awareness and inter-agent coordination.

## 5 RELATED WORK

**Reasoning and planning with LLM agents.** The rise of LLMs has enabled applications in both single- and multi-agent settings. The works in a single-agent setting focus on improving reasoning through chain-of-thought prompting (Wei et al., 2022; Kojima et al., 2022), self-consistency (Wang et al., 2022), and problem decomposition (Zhou et al., 2022). LLMs have also been applied to robotic planning (Ahn et al., 2022), integrated reasoning and acting, and reflection-based learning (Shinn et al., 2023). Zhu et al. (2024) and Qiao et al. (2024) leverage the memory of past actions and states to improve planning. In contrast, our work targets multi-agent environments. In multi-

LLM agent research, Park et al. (2023) proposed a fully automated cooperative framework through manually designed perception, communication, and planning modules, while Li et al. (2023a) facilitates agent communication through role-playing and inception prompting. More recently, data-driven enhancement approaches, such as ReAd (Zhang et al., 2025) refine LLM-generated plans via advantage-weighted action selection. ReAd (Zhang et al., 2025) operates purely in the action–reward space: collects offline trajectories, estimates action advantages, and biases the LLM toward higher-advantage candidate actions during decoding. Our method, in contrast, is designed to extract and enforce a causal dependency structure between past actions and current states of both agents. Thus, while ReAd (Zhang et al., 2025) improves planning through reward-driven preference shaping, CausalPlan improves planning through explicit modeling and enforcement of causal dependencies between agents.

**Zero-shot multi-agent coordination.** Zero-shot multi-agent coordination aims to train agents that can collaborate with unseen partners, human or AI. A classic method is Self-Play (SP) (Tesauro, 1994; Carroll et al., 2019), where agents train by interacting with themselves. Population-Based Training (PBT) (Jaderberg et al., 2017) promotes learning by diversifying the population of training agents. Recent methods combine SP and PBT to increase diversity, such as Fictitious Co-Play (FCP) (Strouse et al., 2021) and Maximum Entropy Population (MEP) (Zhao et al., 2023a). COLE (Li et al., 2023b) shifts focus to strategic policy selection during training. However, these methods are generally computationally expensive and lack interpretability. Zhang et al. (2024a) shows that LLM-based agents can excel in zero-shot tasks by using rich language knowledge. Although this demonstrates the potential of language-based agents, LLMs tend to select causally invalid actions (Gao et al., 2023). To address this challenge, we propose a causal align planning approach that enhances action selection for LLMs.

**Causality in decision making.** Causality has received increasing attention for improving AI decision-making. In single-agent domains, counterfactual methods are used for data augmentation (Pitis et al., 2020; 2022). Corcoll & Vicente (2020) leverage causality to construct variable hierarchies. Zhang et al. (2023b) redistribute rewards based on causal impact. Seitzer et al. (2021) incorporate causal signals into reward shaping. Peng et al. (2022) learns causal graphs to define hierarchical RL subgoals. More recently, efforts have focused on integrating causality into LLM planning by directly providing the causal graph as part of the LLM prompt (Chen et al., 2025; Yu & Lu, 2025). However, a limitation of these approaches is the reliance on the causal reasoning and inference ability of the LLM, which can vary significantly between models and prompts. In multi-agent settings, social influence has been used as causality to promote cooperation (Jaques et al., 2019), while subsequent work employs action influence and redistribution of rewards to encourage coordinated behaviors (Du et al., 2024; Zhang et al., 2024b). In contrast to these lines of research, our work integrates causal modeling directly into multi-agent systems built on LLMs. CausalMACE (Chai et al., 2025) also targets multi-LLM-agent collaboration, prompting an LLM to infer a causal graph from task descriptions and rules, and was designed specifically for Minecraft gameplay. However, our framework CausalPlan does not ask the LLM to construct or infer a causal graph, and is not designed for any specific environment, making it more generally applicable.

## 6    CONCLUSION AND FUTURE WORKS

In this paper, we introduce CausalPlan, a framework designed to integrate causal knowledge into the decoding processes of LLM agents to enhance their performance in multi-agent cooperation. While our SCA model is trained using trajectories from a single behavior policy, our experiments show that it generalizes effectively when deployed with other partner agents, demonstrating robustness in multi-agent coordination. Moreover, we evaluate CausalPlan across multiple environment layouts, confirming that the learned policy-level causal structure remains meaningful and beneficial in different spatial configurations.

This work represents an important step toward incorporating causal knowledge into multi-agent planning with LLMs. Although the framework is not currently intended for deployment in specific applications, it has potential to improve the safety, efficiency, and interpretability of collaborative AI systems. As a promising direction for future work, our approach could be combined with causal prompting methods or layout-agnostic SCA learning to further strengthen planning performance and enable more generalizable multi-agent coordination.

## REPRODUCIBILITY STATEMENT

All implementation details, experimental settings, results are provided and can be found in the Appendix to ensure full reproducibility. The complete source code is also submitted with the submission.

## LLM USAGE

Large Language Models (LLMs) were employed as the backbone for experiments with our CausalPlan framework. We also use LLM to refine the paper's presentation by improving grammar and overall writing clarity.

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

APPENDIX

# A    IDENTIFIABILITY ANALYSIS

Proposition 1 (Identifiability of the causal structure and functions):

Let the dataset consist of sequences of the form:

$$(s_t, a_{t-1}, a_t), \quad t = 1, \ldots, T, \tag{8}$$

where the state $s_t$ and previous action $a_{t-1}$ are observable, while the next action $a_t$ is observable during training and unobservable during inference. Assume the data comes from a Markov Decision Process (MDP) under the interaction of a fixed behavior policy $\pi_\beta$. The next action $a_t$ (a binary vector of size $A$) is assumed to be generated by a structural causal model (SCM):

$$a_{i,t} = f_i(\mathrm{Pa}(a_{i,t})) + \varepsilon_{a_i}, \quad i = 1, \ldots, A, \tag{9}$$

where, the parents $\mathrm{Pa}(a_{i,t})$ of $a_{i,t}$ are selected from the state $s_t$ and action $a_{t-1}$. The noise terms, $\varepsilon$, are independent of the parents. Under the following assumptions:

1. **Additive noise**: The noise terms are independent and identically distributed (i.i.d.) and do not depend on the inputs (Hoyer et al., 2008).

2. **Causal sufficiency**: All relevant causes are observed (i.e., no hidden confounders) (Spirtes et al., 2000).

3. **Faithfulness and global Markov condition**: Observed conditional independencies match those implied by the graph (Pearl, 2009).

4. **Function class expressiveness**: Each function $f_i$ belongs to a class identifiable under additive noise models. In additive noise models, identifiability of causal direction relies on the function class having sufficient expressiveness and satisfying certain regularity conditions (e.g., nonlinearity, invertibility) (Ke et al., 2019; Peters et al., 2017).

5. **Acyclicity**: The causal graph has no cycles (i.e., it is a Directed Acyclic Graph (DAG)).

6. **Sufficient data**: There are enough samples to guarantee reliable estimation.

Then, both the structure of the causal graph and the functions $f_i$ can be identified. In particular, the binary adjacency masks indicating causal edges can be consistently estimated.

**Proof sketch:**

*Step 1: Identifiability using additive noise models.* Under the above assumptions, especially additivity and faithfulness, each causal function $f_i$ can be learned uniquely up to Markov equivalence. Prior work (Hoyer et al., 2008) shows that additive noise and independence of noise from inputs imply identifiability of the direction of causality.

*Step 2: Estimating the functions.* We approximate each function $f_i$ using a weighted basis expansion:

$$f_i(\cdot) \approx W_i^\top \phi_i(\cdot), \tag{10}$$

where $\phi_i(\cdot) \in \mathbb{R}^d$ is a nonlinear feature map that transforms the input tuple $(\cdot)$ into a $d$-dimensional representation, and $W_i \in \mathbb{R}^d$ is the corresponding weight vector of function $f_i$. In the simplest case, $\phi_i(\cdot)$ and $W_i$ are predefined basis functions and linear coefficients, respectively. However, in practice, we often implement $f_i$ using a neural network to allow for flexible function approximation. Given an input tuple $(s_t, a_{t-1})$, the generating function for $a_{i,t}$ can be rewritten as:

$$a_{i,t} = f_i(s_t, a_{t-1}) + \varepsilon_{a_i} \approx W_i^\top \phi_i(s_t, a_{t-1}) + \varepsilon_{a_i}, \tag{11}$$

with noise term $\varepsilon_{a_i}$. Suppose we have a dataset comprising $N$ trajectories $k$ with the form given in Eq. 8. For each trajectory, we define:

$$\Phi_i^k = \begin{bmatrix} \phi_i(s_1^k, a_0^k)^\top \\ \vdots \\ \phi_i(s_T^k, a_{T-1}^k)^\top \end{bmatrix} \in \mathbb{R}^{T \times d}, \qquad \mathbf{A}_i^k = \begin{bmatrix} a_{i,1}^k \\ \vdots \\ a_{i,T}^k \end{bmatrix} \in \mathbb{R}^T, \tag{12}$$

Each row of $\Phi_i^k$ represents the feature vector for a specific time step, while the corresponding element in $\mathbf{A}_i^k$ contains the observed action component. We then estimate $W_i$ by minimizing the ridge-regularized least-squares objective:

$$\min_{W_i} \sum_{k=1}^N \left\| \mathbf{A}_i^k - \Phi_i^k W_i \right\|_2^2 + \lambda \left\| W_i \right\|_2^2, \tag{13}$$

where $\lambda > 0$ controls the regularization strength.

*Step 3: Unique closed-form solution proof.* We proceed by proving that solving the objective yields a unique closed-form solution. The objective function above can be compactly written as:

$$L(W_i) = \sum_{k=1}^N \left\| \mathbf{A}_i^k - \Phi_i^k W_i \right\|_2^2 + \lambda \left\| W_i \right\|_2^2 = \left\| \mathbf{A}_i - \Phi_i W_i \right\|_2^2 + \lambda W_i^\top W_i. \tag{14}$$

First, expand the squared-error term:

$$\left\| \mathbf{A}_i - \Phi_i W_i \right\|_2^2 = (\mathbf{A}_i - \Phi_i W_i)^\top (\mathbf{A}_i - \Phi_i W_i) = \mathbf{A}_i^\top \mathbf{A}_i - 2 W_i^\top \Phi_i^\top \mathbf{A}_i + W_i^\top \Phi_i^\top \Phi_i W_i. \tag{15}$$

Thus,

$$L(W_i) = \mathbf{A}_i^\top \mathbf{A}_i - 2 W_i^\top \Phi_i^\top \mathbf{A}_i + W_i^\top \Phi_i^\top \Phi_i W_i + \lambda W_i^\top W_i. \tag{16}$$

Taking the gradient with respect to $W_i$ gives:

$$\nabla_{W_i} L(W_i) = -2 \Phi_i^\top \mathbf{A}_i + 2 \left( \Phi_i^\top \Phi_i + \lambda I \right) W_i. \tag{17}$$

Setting $\nabla_{W_i} L(W_i) = 0$ yields the normal equation:

$$\left( \Phi_i^\top \Phi_i + \lambda I \right) W_i = \Phi_i^\top \mathbf{A}_i. \tag{18}$$

Since $\lambda > 0$, the matrix $\Phi_i^\top \Phi_i + \lambda I$ is strictly positive-definite and hence invertible. Therefore, the unique minimizer is:

$$\boxed{W_i = \left( \Phi_i^\top \Phi_i + \lambda I \right)^{-1} \Phi_i^\top \mathbf{A}_i.} \tag{19}$$

Re-expressing in terms of the individual trajectories,

$$\Phi_i^\top \Phi_i = \sum_{k=1}^N (\Phi_i^k)^\top \Phi_i^k, \quad \Phi_i^\top \mathbf{A}_i = \sum_{k=1}^N (\Phi_i^k)^\top \mathbf{A}_i^k, \tag{20}$$

so equivalently

$$W_i = \left( \sum_{k=1}^N (\Phi_i^k)^\top \Phi_i^k + \lambda I \right)^{-1} \sum_{k=1}^N (\Phi_i^k)^\top \mathbf{A}_i^k. \tag{21}$$

Because $L(W_i)$ is strictly convex, it admits a unique closed-form solution. Moreover, given a sufficiently large dataset, the estimator converges to a good estimate of $W_i$ Williams & Rasmussen (2006).

To recover the graph structure, we exploit the closed-form solution for $W_i$ derived by minimizing the regularized quadratic loss in the previous step:

$$W_i = \left( \Phi_i^\top \Phi_i + \lambda I \right)^{-1} \Phi_i^\top \mathbf{A}_i.$$

This expression yields an estimate of the weight vector $W_i$, which quantifies the linear relationship between the current state and previous action features and the target component $a_{i,t}$. The *support* of $W_i$—i.e., the indices of its nonzero entries—identifies which features are informative for predicting $a_{i,t}$. Under the *faithfulness* assumption, this support exactly corresponds to the true parent set of node $i$ in the underlying causal graph. Thus, one can recover the graph structure by examining which entries of $W_i$ are significantly nonzero, using thresholding or statistical tests.

CONCLUSION

Under the usual identifiability conditions, both the graph structure and the functional relationships in the Structural Causal Action model are uniquely determined. As a result, the Structural Causal Action model learned by CausalPlan captures a sparse pattern of causal dependencies at the policy-level, providing a stable and interpretable approximation of the underlying decision-making process.

# B  CONCEPTUAL COMPARISON OF RIDGE REGRESSION AND BERNOULLI NLL

In the above Sect. A, the proof uses ridge regression over fixed basis features to simplify the analysis. Here, we clarify the connection to the actual neural Bernoulli heads used in CausalPlan and the assumptions underlying the simplified proof.

## B.1  LOSS FUNCTIONS

- **Ridge regression (with L2 regularization):**

$$\min_w \sum_{i=1}^{N} \left\| y_i - x_i^\top w \right\|_2^2 \ + \ \lambda \left\| w \right\|_2^2, \tag{22}$$

- **Negative log-likelihood (NLL) (with L2 regularization):**

$$-\sum_{i=1}^{N} \left[ y_i \log \sigma(x_i^\top w) + (1 - y_i) \log(1 - \sigma(x_i^\top w)) \right] \ + \ \lambda \|w\|_2^2, \tag{23}$$

## B.2  GRADIENTS

$$\nabla_w \mathcal{L}_{\text{ridge}} = -\sum_{i=1}^{N} (y_i - x_i^\top w)\, x_i \ + \ 2\lambda w, \tag{24}$$

$$\nabla_w \mathcal{L}_{\text{NLL}} = \sum_{i=1}^{N} (\hat{y}_i - y_i)\, x_i \ + \ 2\lambda w, \quad \hat{y}_i = \sigma(x_i^\top w) \tag{25}$$

Here, $y_i$ denotes the ground-truth and $\hat{y}_i$ denotes the prediction.

## B.3  APPROXIMATION

When the predictions are close to the targets ($\hat{y}_i \approx y_i$), the sigmoid function is approximately linear in the neighborhood of the current logit. That is, for small prediction errors, $\hat{y}_i - y_i \approx c\,(x_i^\top w - y_i)$ for some constant $c > 0$. Under this approximation, the NLL gradient resembles the ridge regression gradient up to a scaling factor.

In practice, the actual CausalPlan model uses nonlinear neural Bernoulli heads. The proof in Sect A serves as a **simplified surrogate illustration** rather than a formal identifiability guarantee for the trained model.

## C    CAUSALPLAN DETAILS

As described in Sect. 3 and in Fig. 2, the CausalPlan framework involves two phases Causal Action Structure Learning and Agent Planning with Causal Knowledge. Here, we discuss in detail the two phases and their components, as well as present algorithms that outline the method.

### C.1    CAUSAL ACTION STRUCTURE LEARNING DETAILS

This appendix outlines the procedure used to model and learn the causal relationships between the previous action $a_{t-1}$, current state $s_t$, and next action $a_t$.

**Buffer $B$ Collection.** The data collection process begins by constructing the buffer $B$, which is used to train the SCA model. We collect this data by allowing a pretrained agent to interact with the environment for $N$ timesteps, with each episode having a horizon of $T$. These interactions include both high-level task-oriented actions and low-level movement actions.

To facilitate causal analysis, we apply a preprocessing step in which all low-level movement actions are relabeled as the most recent preceding high-level action of interest. For example, if the agent executes `pickup_onion`, then moves for several steps, and finally performs `put_onion_in_pot`, all intermediate movement actions are relabeled as `pickup_onion`. This yields a simplified sequence: `pickup_onion → pickup_onion → pickup_onion → put_onion_in_pot`. This transformation reduces noise from irrelevant actions and makes it easier to detect meaningful causal edges—such as from `pickup_onion` to `put_onion_in_pot`.

Importantly, we retain the original state observations at each timestep, even after relabeling the actions. This ensures that we can still study the causal relationship between the immediate state before an action and the subsequent high-level decision, preserving the integrity of the underlying state-action dynamics.

**SCA Model.** To capture these dependencies, we employ the SCA model, which incorporates two key components: the generative parameters $\delta$ and the structural parameters $\eta$. The parameters $\delta$ define a set of functions $f$, each implemented as a neural network. Specifically, for each action feature $a_i$ in Eq. 1, there is a corresponding function $f_i$ parameterized by $\delta_i$ (see Appx. D.12 for network details). As described in Sect. 3.1, the model generates the next action $a_t$ based on the current state and previous action. The full training procedure is summarized in Algorithm 1, which alternates between updating the generative $\delta$ and structural parameters $\eta$ using mini-batches sampled from the buffer $B$.

In phase one of the optimization, the parameters $\delta$ govern the generative mapping and are optimized while parameter $\eta$ is fixed. To optimize $\delta$, we draw graph configurations from a Bernoulli distribution based on current edge beliefs: $\mathcal{G}^{sampled} \sim \text{Ber}(\sigma(\eta))$. Each sampled graph acts like a hard intervention, specifying which edges are active, and $\delta$ is trained under these causal hypotheses. Only those factorizations that are parents of the current action feature $a_i$, according to the sampled graph, are activated. We implement this by masking out all features not connected to $a_i$, ensuring that each function $f_i$ conditions only on its relevant causal parents, as defined in Eq. 1. Furthermore, we manually set the diagonal entries of $\mathcal{G}^{sampled}$ to 0, since edges from an action factorization to itself are not allowed in the causal graph. This constraint prevents self-causation among action nodes, maintaining a valid causal structure. The optimization process encourages the model to learn dynamics robust across plausible causal structures.

In phase 2, the structural parameters $\eta$, where each entry indicates the presence or absence of a directed edge between action factorizations, using binary adjacency indicators, is optimized. With the generative parameters $\delta$ fixed, we now update $\eta$ via backpropagation. We applied a sigmoid $\sigma$ function to each entry of $\eta$ producing values in range $[0, 1]$ that represent the probability of an edge's existence and mask the features not connected to the current feature $a_i$ using this soft-intervention. Here, we also manually set the diagonal entries of $\eta_{i \to i} = 0$, since edges from an action factorization to itself are not allowed in the causal graph. Essentially, we are refining the graph toward the structure best supported by the data. Edges that improve predictions are reinforced, while edges that hurt performance are down-weighted .

**Optimization.** The overall loss function is defined as
$$L(\delta, \eta) = L_{\text{causal}}(\delta, \eta) + L_{\text{reg}}(\eta),$$

where $L_{\text{causal}}$ encourages accurate prediction of the next action, and $L_{\text{reg}}$ regularizes the structural parameters to promote sparsity and prevent overfitting. This results in an interpretable and reliable causal model. Refer to (Ke et al., 2019) for details of this two-phase optimization process, which we adapt in our method.

---

**Algorithm 1** Iterative Optimization for Structural Causal Action (SCA) Model with Bernoulli Sampling

---

1: **Input:** Dataset $B = \left\{ \{(s_t^k, a_t^k)\}_{t=1}^T \right\}_{k=1}^N$
2: Initialize structural parameters $\eta$ and generating parameters $\delta$
3: **Repeat:**
4:    **1. Sample a mini-batch** $\mathcal{B} = \left\{ \{(a_{t-1}^k, s_t^k, a_t^k)_{t \in T}\} \right\}_{k \in N} \subset B$
5:    **2. Optimize Generating Parameters** $\delta$**:**
6:       **Fix** $\eta$
7:       **For each forward pass:** sample a graph $\mathcal{G}^{sampled} \sim \text{Ber}(\sigma(\eta))$
8:       Mask edges according to sampled $\mathcal{G}^{sampled}$
9:       Optimize $\delta$ by minimizing the loss $L_{\text{causal}}(\delta, \eta)$ in Eq. 2
10:      Update generating parameters $\delta$
11:    **3. Optimize Structural Parameters** $\eta$**:**
12:      **Fix** $\delta$
13:      **Apply** $\sigma(\eta)$
14:      Update $\eta$ via backpropagation using loss $L(\delta, \eta) = L_{\text{causal}}(\delta, \eta) + L_{\text{reg}}(\eta)$ (Eq. 2, Eq. 3)
15: **Output:** Optimized parameters $\delta, \eta$

---

### C.1.1 STATE AND ACTION FACTORIZATION

We assumed a known factorization of state and action spaces, a common assumption often made in causal reinforcement learning research (Seitzer et al., 2021; Peng et al., 2022). This allows us to encode the states and actions into binary vectors: $s_t = [s_{t,1}, \ldots, s_{t,S}] \in \{0,1\}^S$, $a_t = [a_{t,1}, \ldots, a_{t,A}] \in \{0,1\}^A$, where each component $s_{t,j}$ and $a_{t,i}$ is a binary indicator representing whether a particular state feature or action is active (1) or inactive (0).

For example, given an observation $s_t^k$ of trajectory $k$ at timestep $t$: "agent 1 is holding an onion, agent 2 is holding nothing", this can be encoded into a binary state vector such as:

$$s_t^k = [1, 0, 0, 1, 0, \ldots],$$

where each entry corresponds to a specific feature (e.g., "agent 1 is holding onion", "agent 1 is holding nothing", "agent 2 is holding nothing", etc.), and the 1s indicate which conditions are currently true. Similarly, an action like "agent 1 places onion in pot" can be encoded into

$$a_t^k = [0, 1, 0, \ldots],$$

where each entry corresponds to a specific atomic action in the action space, and the 1 marks the active action at time $t$. **Note:** In our training process, we use only the previous action of the controlling agent. While it is possible to incorporate the actions of the other agent, doing so increases the complexity of learning the causal graph and may negatively impact the training performance.

This factorized representation enables us to formulate the causality training as a classification problem, allowing us to optimize using the negative log-likelihood loss defined in Eq. 2. Refer to Appx. D.12.2 for the factorization features used in our experiments.

In practice, the assumption of discrete variables and state-action factorization do not restrict the broader applicability of CausalPlan. Many practical domains—such as recommendation systems and text-based planning agents—naturally produce structured, vector-based observations that can be discretized or directly mapped to symbolic variables. Our evaluation closely reflects real-world scenarios such as Model Context Protocol (MCP) servers, where agents receive textual prompts and select appropriate API function calls. Similarly, text-based recommendation systems process natural language inputs about user preferences to generate textual suggestions. These examples share the symbolic, text-in/text-out framework of Overcooked-AI, underscoring the practical relevance and generalizability of our approach beyond simulated settings. Additionally, a number of methods

have been developed to recover such symbolic factors from high-dimensional inputs for the purpose of causal discovery: for example, CausalVAE (Yang et al., 2021), DEAR (Shen et al., 2020), and more recently VLM-based approaches learn disentangled latent state-action representations, while VACERL (Nguyen et al., 2024) demonstrates effective causal discovery directly in image-based environments. The main limitation in moving to continuous space lies in the complexity of learning a mapping function that must be learned during causal modeling.

## C.2    AGENT PLANNING WITH CAUSAL KNOWLEDGE DETAILS

This appendix provides additional details on how causal knowledge is integrated into the agent's decision-making process during action planning.

**LLM prompting process.** During inference, we first equip the LLM agents with a knowledge library that specifies the tasks, rules, and example responses relevant to the game environment. At each time step, the current observation $s_t$ is presented to the agent along with a prompt instructing it to analyze the situation. The agent typically responds with a natural language interpretation highlighting the key elements of the observation. Both the original observation $s_t$ and the generated analysis are then fed into a second prompt, which instructs the agent to produce a set of appropriate next actions $\mathcal{A}'$. For further details, refer to Appx. C.2.1.

**Causal-Aware Planning.** When a set of candidate actions $\mathcal{A}'$ is generated during planning, each action is initially assigned a probability by the LLM model, denoted as $P_a(\mathcal{A}')$. To incorporate causal reasoning, the agent queries the Causal Action Matrix $\mathcal{M}$ using the current state $s_t$ and previous action $a_{t-1}$ to compute a corresponding set of causal scores $P_c(\mathcal{A}')$ (refer to Appx. C.2.2 for details). A weighted combination of the LLM's probabilities and causal scores is formed using Eq. 4 and then normalized via the softmax function:

$$\frac{\exp(p_f(a'_m))}{\sum_{j=1}^{|\mathcal{A}'|} \exp(p_f(a'_m))}, \tag{26}$$

resulting in the final action distribution $P_f(\mathcal{A}')$. Redundant actions are identified and merged according to the process described in Appx. C.2.4, and the agent samples the next action $a_{t+1}$ from this refined distribution.

**Causal Backup Plan.** In scenarios where no valid candidate actions are proposed (i.e., $\mathcal{A}' = \emptyset$), mostly due to hallucinations, the agent relies on a causal fallback mechanism. Instead of halting execution, it queries $\mathcal{M}$ using $s_t$ and $a_{t-1}$ to derive a causal distribution over the original instruction set $\mathcal{A}$. The agent then selects the action with the highest causal score, effectively leveraging prior experience to recover from failure.

The complete inference procedure using Causal-Aware Planning and Causal Backup Plan is summarized in Algorithm 2.

### C.2.1    LLM PROMPT DESIGN

**Knowledge library.** At the beginning of the inference process, we construct a knowledge library for the LLM agent, following prior work in the field (Zhang et al., 2024a; Qiao et al., 2024). This library is organized around three key perspectives: the tasks, the rules, and the in-context examples. This knowledge library is fed into the LLMs at the initial stage of the inference process before the cooperation task begins. An example of a knowledge library is provided in Fig. 5.

In our experiments, for simplicity, we utilized the knowledge library provided by Zhang et al. (2024a), with slight modifications to accommodate our two-prompt design, as their work uses the same evaluation environment[1].

---

[1]https://github.com/PKU-Alignment/ProAgent (MIT License).

---

**Algorithm 2** Agent Planning with the Causal Knowledge Algorithm at time step $t$

---

1: **Input:** Current state $s_t$, previous action $a_{t-1}$, candidate actions $\mathcal{A}'$, LLM probabilities $P_a(\mathcal{A}')$, instruction set $\mathcal{A}$, causal matrix $\mathcal{M}$, weighting coefficient $\gamma \in [0,1]$, $P_f(\mathcal{A}') = \emptyset$, $P_c(\mathcal{A}) = \emptyset$
2: **If** $\mathcal{A}' \neq \emptyset$ **then**
3:     **For all** $a'_m \in \mathcal{A}'$
4:         $p_c(a'_m) \leftarrow \mathcal{M}(s_t, a_{t-1}, a'_m)$
5:         $p_f(a'_m) \leftarrow \gamma \cdot p_a(a'_m) + (1-\gamma) \cdot p_c(a'_m)$ (Eq. 4)
6:         $P_f(\mathcal{A}') \leftarrow p_f(a'_m)$
7:     **End for**
8:     Normalize $P_f(\mathcal{A}')$ using softmax in Eq 26
9:     Apply redundancy check (see Appx. C.2.4) to get $\mathcal{A}'^*$, $P_f^*$
10:    Sample $a_t \sim \text{Categorical}\left( \left[ p_f^*(a'_1), p_f^*(a'_2), \ldots, p_f^*(a'_{|\mathcal{A}'^*|}) \right] \right)$
11: **Else**
12:     **For all** $a \in \mathcal{A}$
13:         $p_c(a) \leftarrow \mathcal{M}(s_t, a_{t-1}, a)$
14:         $P_c(\mathcal{A}) \leftarrow p_c(a)$
15:     **End for**
16:     $a_t \leftarrow \arg\max_{a \in \mathcal{A}} P_c(a)$
17: **End If**
18: **Output:** Selected action $a_t$

---

| **Knowledge library** |
|---|
| *Tasks:* |
| - You are ... |
| - This is a team game played by two players who will ... |
| - The team goal is ... |
| - You need to ... |
|   |
| *Rules:* |
| - In this task, the legal actions include: [Action 1], [Action 2], ... |
| - Assume the role of an assistant proficient in the task. Your objective is to control Player 0 and cooperate with Player 1, who follows a fixed strategy, in order to achieve a high score. You should adhere to the following guidelines: |
| - [Rule 1]. |
| - [Rule 2]. |
| - ... |
|   |
| - For each step, you will receive the current scene or current scene with an analysis. |
| - If you receive only the current scene, you need to: |
|    1. Describe the current scene and analyze it. |
| - If you receive the current scene and the analysis then you need to: |
|   2. Plan ONLY ONE best skill for   to do right now. Format should be ... |
|   |
| *Examples:* |
| ### |
| Scene 1 Prompt 1: [Environment Scene 1][Player 0 Scene 1] [Player 1 Scene 1]. |
|   |
| Analysis: Both player are [Scene Description]. I believe [Other Analysis]. |
| ### |
| Scene 1 Prompt 2: [Environment Scene 1][Player 0 Scene 1] [Player 1 Scene 1]. Analysis: Both Player are [Scene Description]. I believe [Other Analysis]. |
|   |
| Plan: Player 1 should [Scene 1 Action]. |
| ### |
| Scene 90 Prompt 1: [Environment Scene 90][Player 0 Scene 90] [Player 1 Scene 90]. |
|   |
| Analysis: Player 0 and Player 1 are [Scene 90 Description]. I believe [Other Analysis]. |
| ### |
| Scene 2 Prompt 2: [Environment Scene 90][Player 0 Scene 90] [Player 1 Scene 90]. Analysis: Player 0 and Player 1 are [Scene 90 Description]. I believe [Other Analysis]. |
|   |
| Plan: Player 1 should [Scene 90 Action]. |
| ### |
| ... |

Figure 5: An Example of Knowledge Library.

**Analysis and planning prompts** To facilitate the planning process, we first ground the environment state into natural language so that it becomes interpretable to the LLM agent, as the raw state representation is typically not directly understandable by language models. In our experiments, we adopt the grounding methodology proposed by Zhang et al. (2024a), since their work uses the same evaluation environments. For detailed grounding procedures, we refer the reader to their paper. An example of the final grounded state prompt used as input to the agent at each timestep is highlighted in red in Fig. 6.

We then apply our two-prompt design to guide the LLM's behavior using the knowledge library. Specifically, when the agent is prompted with only the current observation, it is expected to analyze the scene. When the prompt includes both the observation and the analysis, the agent is expected to respond with a planned action. Our approach first asks the agent to perform the analysis, then uses that analysis together with the state prompt as input to generate the final action plan. The analysis is highlighted in green, while the planned action is highlighted in purple in Fig. 6. We hypothesize that this two-prompt process provides the agent with a reasoning workflow similar to the chain-of-thought (CoT) prompting described by Wang et al. (2022), while also allowing straightforward access to the planned action through hard-coded separation. In contrast, including both the analysis and the planned action in the same response, as done by Zhang et al. (2024a)—can make it difficult to accurately extract the planned action, since action names might appear within the analysis. We evaluate the performance of one-prompt versus two-prompt approaches without causality enhancement through our CausalPlan in Sect. 4.4 and find that the results are quite similar, with the two-prompt approach showing slightly better performance. Although the single-prompt approach is feasible in practice, it complicates reliably identifying the correct action.

---

**Analysis and planning prompts**

###
Scene 1 Prompt 1: Layout: Onion Dispenser 0, Onion Dispenser 1, Dish Dispenser 0, Serving Location 0, Pot 0, Pot 1.
State: Player 1 holds nothing. Player 0 holds nothing. Kitchen states: Pot 0 is empty. Pot 1 is empty. 3 counters can be visited by Player 0. Their states are as follows: No counters have onion. No counters have dish.

Analysis:
- The pot is empty, and there are no onions or dishes on the counters.
- Player 1 needs to deliver an onion and a dish to the counter.
- The legal actions that player1 can take are pickup(onion) and pickup(dish).
###
Scene 1 Prompt 2: Layout: Onion Dispenser 0, Onion Dispenser 1, Dish Dispenser 0, Serving Location 0, Pot 0, Pot 1.
State: Player 1 holds nothing. Player 0 holds nothing. Kitchen states: Pot 0 is empty. Pot 1 is empty. 3 counters can be visited by Player 0. Their states are as follows: No counters have onion. No counters have dish. Analysis:
- The pot is empty, and there are no onions or dishes on the counters.
- Player 1 needs to deliver an onion and a dish to the counter.
- The legal actions that player1 can take are pickup(onion) and pickup(dish).

Plan: Player 1 should pickup(onion).
###

Figure 6: An example of analysis and planning prompts.

### C.2.2 CAUSAL KNOWLEDGE CONSULTATION DETAILS

To compute the causal score for a candidate action, the agent first maps the action to its corresponding row in $\mathcal{M}$ and identifies which columns are currently active based on features derived from the current state and previous action. These active features are determined using the procedure outlined in Appx. C.2.3.

For instance, given that we want to extract the causal scores of an action $a$, given current state $s_t$ and previous action $a_{t-1}$, we first identify the corresponding index $i$ of the action $a$ within the matrix row. Let $\mathbf{idx} \colon \mathcal{A} \to \{1, \ldots, |\mathcal{A}|\}$ be the function that maps any action to its row index in $\mathcal{M}$, and let $J = \mathrm{Active}(s_t, a_{t-1}) \subseteq \{1, \ldots, S + A\}$ denote the set of column indices corresponding to the features that are "active" in the current state $s_t$ and the previous action $a_{t-1}$.

For a candidate action $a$, we first compute its row index $i = \mathbf{idx}(a)$, then gather the entries of row $i$ in $\mathcal{M}$ at all active columns $j \in J$, thus a query $\mathcal{M}(s_t, a_{t-1}, a)$ will return:

$$p_c(a) = \sum_{j \in J} \eta_{ji}. \tag{27}$$

In other words, $p_c(a)$ is the sum of the causal-weight entries in the row for $a_t$ that correspond to the features currently active.

### C.2.3 EXTRACTING INFORMATION FOR CAUSAL KNOWLEDGE CONSULTATION

Given the observations grounded in natural language, as explained in Appx. C.2.1, we map them to a set of predefined state features. For example, from the state prompt shown in Fig. 7 — "Player 1 holds nothing. Player 0 holds nothing. Kitchen states: Pot 0 is empty. Pot 1 has 1 onion..." — we extract factorized features such as `hold_nothing1` (indicating that agent 1 is holding nothing), `hold_nothing2` (agent 2 is holding nothing), `pot0_0` (pot 0 is empty), `pot1_1` (pot 1 has 1 onion). This allows us to formulate the state feature factorization $s_t$.

In addition, the previous action taken by the agent is also recorded, referring to the last executable action performed. This allows us to configure the action feature factorization vector $a_{t-1}$.

Depending on the environment, the number of factorized features can vary widely (see Appx. D.12.2 for the specific factorized features used in each experimental task). While a larger number of features can produce a more detailed causal graph, this does not necessarily lead to better performance, as learning such graphs becomes more challenging and requires more data. In our experiments, we chose to use high-level state and action factorizations (we ignore low-level movement actions and only focus on the state of the two agents and the pot) to strike a balance between expressiveness and learnability.

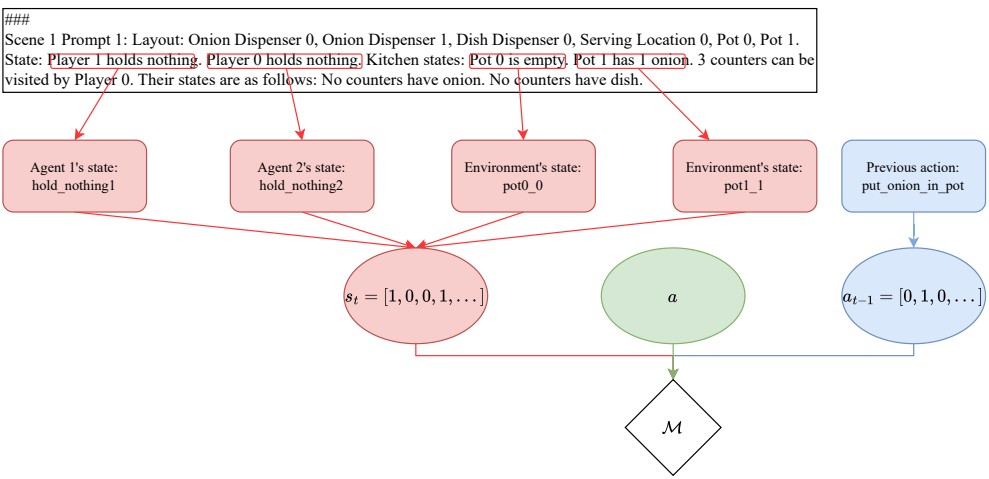

Figure 7: Information extraction for causal knowledge consultation.

### C.2.4 POST-PROCESSING TO IDENTIFY REDUNDANT ACTIONS

During the process of sampling the next actions, the LLM may output the same action in different formats within the sampled set $\mathcal{A}'$. To address this, we apply a series of post-processing steps using standard natural language processing techniques—such as converting text to lower-case, removing punctuation, and regex matching pre-defined patterns—to identify and merge semantically equivalent actions. This enables us to accurately aggregate their probabilities in $P_f(\mathcal{A}')$. For instance, the same action `put_onion_in_pot` can be expressed as `put_onion_in_pot()`, `put_onion_in_pot().`, or `put_onion_In_Pot` (refer to the associated code for details of this process). After post-processing all these possible responses, we can calculate the updated value:

$$p_{\mathrm{f}}(\texttt{put\_onion\_in\_pot}) = p_{\mathrm{f}}(\texttt{put\_onion\_in\_pot()})$$
$$+ \, p_{\mathrm{f}}(\texttt{put\_onion\_in\_pot().}) + p_{\mathrm{f}}(\texttt{put\_onion\_In\_Pot})$$

## D ADDITIONAL EXPERIMENT DETAILS

### D.1 CAUSALPLAN IMPLEMENTATION

As mentioned earlier, we build upon the ProAgent framework (Zhang et al., 2024a), retaining all components except for the planning module, which we replace with our proposed algorithm. Unlike the original ProAgent implementation that relied on the closed-source `GPT-3.5` for planning, we instead utilize one of the following open-source language models, all retrieved from Hugging Face[2]: `gemma-1.1-7b-it` (Gemma-7B), `Meta-Llama-3-8B-Instruct` (Llama-8B), `Qwen2.5-14B-Instruct-1M` (Qwen-14B), and `Llama-3.3-70B-Instruct` (Llama-70B). These models are integrated into the ProAgent framework to serve as the core planner, with our `CausalPlan` method applied to refine the generated actions. Additionally, for the two-prompt input structure, we employ the `Cohere/command-r` model (Cohere, 2024)—a 35-billion-parameter LLM accessed via the Cohere API using the official `cohere` Python client[3]—to produce scenario analyses for faster inference. For the "Belief Correction" module, we also substitute `GPT-3.5` with the same Cohere model. The "Controller" module in ProAgent (Zhang et al., 2024a)—and in our setup—uses a rule-based best-first search; while effective, performance could likely be improved with a reinforcement learning-based approach.

Regarding hardware requirements, the Gemma-7B and Llama-8B models each require approximately 10–16 GB of VRAM, Qwen-14B demands around 25–30 GB and multi-GPU support, while Llama-70B needs over 70 GB VRAM with multi-GPU configuration on NVIDIA h-100 GPUs.

To facilitate easier extraction of action selection probabilities, we slightly modify the prompting strategy used in the original method. In particular, we separate the reasoning step, based on CoT prompting, from the action planning step, implementing them as two distinct prompts. The output of the reasoning prompt is then used as input for the planning prompt. We provide further details of this process in Appx. C.2.1 and include an empirical study in Appx. D.7, demonstrating that this modification does not contribute to the performance gains, nor does it substantially affect the overall performance of the backbone.

To avoid the cold-start problem and long interaction times associated with using small LLMs to collect data into the buffer $B$, we employ a pre-trained policy based on MEP to interact with the environment and gather data. Nonetheless, we conduct an experiment (results are in Appx. D.7) demonstrating that even when using a small LLM, specifically Llama-8B, for data collection, our method still yields improved performance compared to simply using the backbone method.

### D.2 ENVIRONMENT DETAILS

We use the Overcooked-AI environment suite as our testing platform (Carroll et al., 2019). In Overcooked, two agents must collaborate to prepare and serve onion soup. Their tasks include gathering and placing up to three ingredients into a pot, cooking the soup, transferring it into a dish, and delivering the final meal. Each successful delivery yields a reward of +20, and both agents share the final return, promoting cooperative behavior. This suite comprises five distinct layouts (Carroll et al., 2019)—*Cramped Room* (CR), *Asymmetric Advantages* (AA), *Coordination Ring* (COR), *Forced Coordination* (FC), and *Counter Circuit* (CC)—each designed to evaluate different aspects of multi-agent collaboration under varying levels of complexity and coordination demands:

- **Cramped Room (CR):** This environment features a highly constrained layout with narrow hallways and tight corridors, forcing agents to navigate around each other constantly.
- **Asymmetric Advantages (AA):** In AA, the kitchen layout provides one agent with easier access to ingredients and tools, while the other agent is disadvantaged in terms of spatial reach.

---

[2]`https://huggingface.co`
[3]`https://docs.cohere.com/v2/reference/chat`

- **Coordination Ring (COR):** COR introduces a ring-like structure in the kitchen, where ingredients, cooking stations, and delivery points are spread along a loop.
- **Forced Coordination (FC):** FC is designed to enforce interdependence between the agents through environment constraints.
- **Counter Circuit (CC):** The CC environment includes a set of counters that create a barrier between the agents and the task stations.

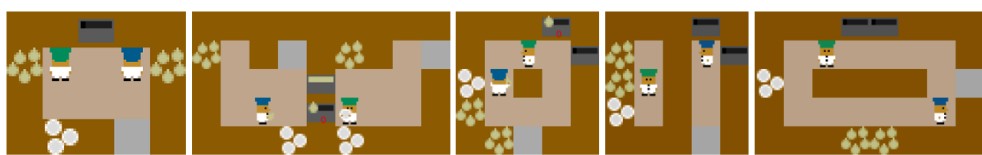

Figure 8: Overcooked-AI Environments. From left to right: Cramped Room (CR), Asymmetric Advantages (AA), Coordination Ring (CR), Forced Coordination (FC), and Counter Circuit (CC).

The environment testing suite was collected from the associated GitHub repository[4].

### D.3 BASELINE DETAILS

We compare CausalPlan against several established reinforcement learning (RL) methods specifically designed for zero-shot human-AI coordination tasks. These baselines have demonstrated strong performance in prior research and serve as competitive benchmarks in our experiments.

- **SP (Self-Play)** (Tesauro, 1994; Carroll et al., 2019): A classical RL approach where agents learn policies by playing against themselves, promoting strategic behavior without relying on external partners.
- **PBT (Population-Based Training)** (Jaderberg et al., 2017): An evolutionary algorithm that optimizes agent populations by iteratively mutating and selecting promising policies, facilitating diverse and robust coordination strategies.
- **FCP (Fictitious Co-Play)** (Strouse et al., 2021): A method that models coordination by simulating the behaviors of various partner types, enabling agents to adapt to unseen collaborators.
- **MEP (Maximum Entropy Population)** (Zhao et al., 2023a): This approach promotes diversity within agent populations by maximizing entropy, which encourages exploration of varied strategies for better coordination.
- **COLE (Cooperative Learning)** (Li et al., 2023b): An algorithm designed to enhance cooperative behavior between agents by explicitly learning to predict and adapt to partners' actions.

These baselines were selected due to their relevance and proven success in multi-agent coordination scenarios. The pretrained baseline models were obtained from the ProAgent GitHub repository[5].

We also evaluate CausalPlan in collaboration with a human policy collected via behavior learning, available at the COLE platform[6].

### D.4 DETAILS OF AI PARTNER EVALUATION

Tab. 4 presents a comprehensive comparison of the performance of various backbone LLMs, both with and without CausalPlan, evaluated across multiple layouts.

---

[4]https://github.com/HumanCompatibleAI/overcooked_ai (MIT License)

[5]https://github.com/PKU-Alignment/ProAgent (MIT License)

[6]https://github.com/liyang619/COLE-Platform

Table 4: Performance of different backbones with and without CausalPlan across various layouts (this is the detailed version of Fig. 3). The reported results, including mean and variance, are obtained from 3 different seeds, with each seed running for 400 timesteps. In these experiments, we use the small LLM agent as Player 1, allowing it to collaborate with all other baselines as described in Sect. 4.1, and report the average and variance of the outcomes. The last column reports the average improvement across backbones, and the last row reports the average improvement across layouts in %. The result with the highest improvement is highlighted in **bold**, while the second highest is underscored.

| Backbones | With CausalPlan | Layouts | | | | | Avg. Improv. (%) |
|---|---|---|---|---|---|---|---|
| | | CR | AA | COR | FC | CC | |
| Gemma-7B | × | $121.3 \pm 16.2$ | $88.0 \pm 32.7$ | $78.7 \pm 8.3$ | $17.3 \pm 6.1$ | $73.3 \pm 10.1$ | 12.82 |
| | ✓ | $141.3 \pm 6.1$ | $122.7 \pm 12.9$ | $82.7 \pm 30.5$ | $17.3 \pm 9.2$ | $78.7 \pm 14.0$ | |
| Llama-8B | × | $110.7 \pm 12.8$ | $163.4 \pm 3.3$ | $80.0 \pm 41.7$ | $9.3 \pm 2.3$ | $84.0 \pm 20.8$ | 13.90 |
| | ✓ | $150.7 \pm 2.3$ | $182.2 \pm 18.3$ | $77.3 \pm 14.0$ | $16.0 \pm 4.0$ | $90.7 \pm 2.3$ | |
| Qwen-14B | × | $117.3 \pm 4.6$ | $224.0 \pm 22.6$ | $76.0 \pm 17.4$ | $16.0 \pm 4.0$ | $48.0 \pm 22.6$ | **29.04** |
| | ✓ | $162.6 \pm 9.2$ | $232.0 \pm 31.7$ | $121.3 \pm 16.6$ | $17.3 \pm 12.8$ | $93.3 \pm 22.7$ | |
| Llama-70B | × | $144.0 \pm 18.3$ | $248.0 \pm 22.7$ | $125.3 \pm 10.0$ | $34.7 \pm 14.0$ | $89.3 \pm 32.3$ | 22.42 |
| | ✓ | $178.7 \pm 2.3$ | $266.7 \pm 16.7$ | $157.3 \pm 2.3$ | $38.7 \pm 16.2$ | $112.0 \pm 6.9$ | |
| Avg. Improv. (%) | – | **20.83** | 18.80 | 19.13 | 4.87 | 9.55 | – |
| Oracle GPT | – | $194.2 \pm 10.5$ | $229.8 \pm 21.9$ | $183.0 \pm 31.7$ | $31.0 \pm 33.9$ | $128.5 \pm 28.1$ | – |

Tab. 5 provides an in-depth comparison between baseline agents and our proposed CausalPlan method using Llama-70B, across different layouts.

Table 5: Performance comparison between baseline agents and CausalPlan (Ours) across layouts using Llama-70B (this is the detailed version of Tab. 1). Results (mean ± variance) are averaged over 3 seeds (400 timesteps each). The first row per layout corresponds to our agent as Player 0, the second to Player 1. Best and second-best results are in **bold** and underlined, respectively.

| Layout | Baseline AI Agents | | | | | CausalPlan (Ours) |
|---|---|---|---|---|---|---|
| | SP | PBT | FCP | MEP | COLE | |
| CR | $160.0 \pm 4.0$ | $165.3 \pm 1.7$ | **$194.6 \pm 10.0$** | $177.3 \pm 22.0$ | $164.0 \pm 6.9$ | $166.7 \pm 6.1$ |
| | $164.0 \pm 16.0$ | $170.7 \pm 8.3$ | **$193.3 \pm 10.1$** | $178.7 \pm 10.1$ | $142.7 \pm 18.0$ | $178.7 \pm 2.3$ |
| AA | $173.3 \pm 22.0$ | $185.3 \pm 12.8$ | $181.3 \pm 14.0$ | $153.3 \pm 2.3$ | $197.3 \pm 14.0$ | **$250.7 \pm 16.1$** |
| | $194.7 \pm 12.9$ | $150.7 \pm 18.0$ | $172.0 \pm 16.0$ | $181.3 \pm 9.2$ | $173.3 \pm 16.2$ | **$266.7 \pm 16.7$** |
| COR | $106.7 \pm 12.8$ | $138.7 \pm 12.2$ | $138.7 \pm 2.3$ | **$166.7 \pm 8.3$** | $154.7 \pm 2.3$ | $156.0 \pm 4.0$ |
| | $134.7 \pm 9.2$ | $140.0 \pm 8.0$ | $122.7 \pm 10.1$ | $154.7 \pm 6.1$ | $152.0 \pm 6.9$ | **$157.3 \pm 2.3$** |
| FC | $10.7 \pm 4.6$ | $20.0 \pm 14.4$ | $57.3 \pm 6.1$ | $22.7 \pm 4.6$ | $41.3 \pm 10.0$ | **$69.1 \pm 13.6$** |
| | $25.3 \pm 4.6$ | **$61.3 \pm 6.1$** | $26.7 \pm 8.3$ | $38.0 \pm 6.1$ | $48.0 \pm 4.0$ | $38.7 \pm 16.1$ |
| CC | $62.7 \pm 12.2$ | $56.0 \pm 8.0$ | $64.0 \pm 8.0$ | $33.3 \pm 22.0$ | $96.0 \pm 4.0$ | **$113.3 \pm 8.3$** |
| | $50.7 \pm 6.1$ | $48.0 \pm 20.0$ | $62.7 \pm 12.9$ | $66.7 \pm 10.1$ | $85.3 \pm 16.2$ | **$112.0 \pm 6.9$** |

## D.5 DETAILS OF HUMAN PARTNER EVALUATION

Our main goal is to present a modular causal reasoning framework that improves LLM-based planning agent that can collaborate well with human.

To provide quantitative support, we present Table 6, which compares Llama-70B with CausalPlan against the best RL baseline and Table 7, which reports the $t$-values of models with and without CausalPlan when paired with a human agent. The $t$-test is a statistical method used to determine whether observed differences between two groups are statistically significant or could have occurred by chance. Higher $t$-values indicate stronger evidence that the difference is meaningful.

**Statistical analysis with best RL methods**

In most environments, CausalPlan leads to a clear improvement in $t$-statistics, often reversing a negative score into a positive one (e.g., CR-P0, AA-P1, FC-P0, CC-P0). Although some $t$-values do not reach statistical significance due to the small sample size ($n = 5$), which is limited by the availability of human data, the consistent trend of improvement suggests that our approach is effective and broadly applicable. We hypothesize that applying our causal method on stronger models like GPT-3.5, as used in ProAgent Zhang et al. (2024a), would likely yield even more significant improvements in performance.

Table 6: $t$-values of models with and without CausalPlan against the best RL baseline.

| Layout | Best RL | $t$-value (w/ CausalPlan) | $t$-value (w/o CausalPlan) |
|--------|---------|---------------------------|----------------------------|
| CR-P0 | COLE | 0.868 | -3.504 |
| CR-P1 | MEP | 0.388 | -1.638 |
| AA-P0 | MEP | 3.329 | 2.660 |
| AA-P1 | COLE | 1.966 | -1.188 |
| COR-P0 | MEP | 0.657 | 0.000 |
| COR-P1 | COLE | 2.781 | 0.838 |
| FC-P0 | COLE | 0.818 | -1.405 |
| FC-P1 | PBT | -0.589 | -1.421 |
| CC-P0 | COLE | 1.099 | -0.211 |
| CC-P1 | COLE | -0.236 | -0.408 |

**Statistical analysis with backbone**

Statistically significant improvements ($p < 0.05$) are observed in 30% of the cases (CR-P0, AA-P1, COR-P1), with strong $t$-values (3.805, 2.987, 2.834 respectively), providing direct evidence that CausalPlan improves performance in these settings. An additional 30% of cases (CR-P1, FC-P0, CC-P0) show marginally significant improvements, with $p$-values between 0.05 and 0.2. These results suggest a positive trend toward significance that may be confirmed with more data. Importantly, 100% of the $t$-values are positive, meaning CausalPlan never degrades performance compared to the non-causal baseline.

Table 7: Paired $t$-test results comparing Llama-70B with CausalPlan (Ours) and Llama-70B (React+Reflexion).

| Layout | $t$-value | $p$-value |
|--------|-----------|-----------|
| CR-P0 | 3.805 | 0.0304 |
| CR-P1 | 1.731 | 0.1982 |
| AA-P0 | 0.490 | 0.6518 |
| AA-P1 | 2.987 | 0.0429 |
| COR-P0 | 0.608 | 0.5867 |
| COR-P1 | 2.834 | 0.0496 |
| FC-P0 | 1.832 | 0.1740 |
| FC-P1 | 0.741 | 0.5000 |
| CC-P0 | 1.902 | 0.1320 |
| CC-P1 | 0.274 | 0.8028 |

## D.6 EFFECT OF HYPERPARAMETER $\gamma$

In our framework, the hyperparameter $\gamma$ in Eq. 10 controls the balance between the agent's belief and the causal knowledge. To investigate the effect of varying $\gamma$, we conducted an experiment on two layouts, CR and FC, using Qwen-14B as the backbone LLM. As shown in Fig. 9, the optimal value for $\gamma$ lies within the range of 0.5 to 0.7. In both cases, when $\gamma$ is set to 0.2, indicating a greater reliance on causal knowledge than on the agent's own knowledge, or when $\gamma$ is set to 1, fully trusting the agent, the performance degraded. Refer to Fig. 9 for the experimental results and Tab. 13 for the $\gamma$ values used for each LLM agent across different layouts. Due to limited computational resources,

tuning was only performed on layouts where CausalPlan initially underperformed with $\gamma = 0.5$. We believe that further tuning of this hyperparameter would likely lead to improved performance.

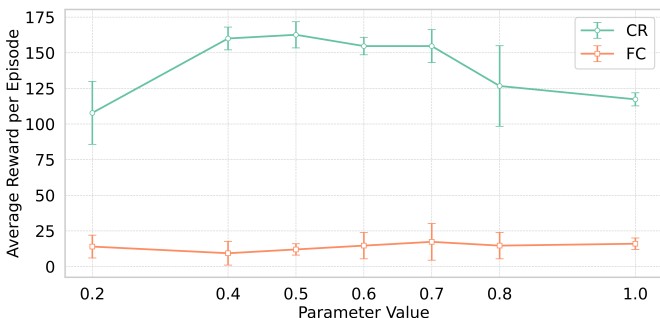

Figure 9: Experiments showing the impact of tuning the hyperparameter $\gamma$ conducted using Qwen-14B on CR and FC layouts. The results, including mean and variance, are averaged over three different seeds. The optimal value of $\gamma$ typically lies within the range of 0.4-0.8, emphasizing the importance of balancing between the belief of the LLMs and the prior causal knowledge.

### D.7 EFFECT OF DIFFERENT DATA COLLECTION POLICY

Table 8: Ablation studies on using different agents to collect data for buffer $B$ conducted on CR layout with Llama-8B as backbone. The results, including mean and variance, are obtained from 3 different seeds. "Llama-8B" and "MEP" refer to using Llama-8B or MEP to generate data.

| Methods | Baseline AI Agents | | | | | Average Results |
|---|---|---|---|---|---|---|
| | SP | PBT | FCP | MEP | COLE | |
| **Llama-8B** | $106.7 \pm 41.6$ | $86.7 \pm 75.1$ | $166.7 \pm 41.6$ | $126.7 \pm 30.6$ | $140.0 \pm 0.0$ | $125.3 \pm 30.7$ |
| **MEP** | $126.7 \pm 30.6$ | $133.3 \pm 30.5$ | $160.0 \pm 40.0$ | $166.7 \pm 41.6$ | $166.7 \pm 23.1$ | $\mathbf{150.7 \pm 2.3}$ |

Tab. 8 presents an ablation study comparing the effects of using different agents—Llama-8B and MEP—for data collection in buffer $B$, when interacting with the environment to collect data for 200k steps. We hypothesize that using MEP for data collection would yield better results, given that it is a pretrained agent specialized for the task. Nevertheless, even when using data collected by Llama-8B, incorporating causal knowledge still provides a performance gain compared to not using causal knowledge at all. The results show that MEP consistently outperforms Llama-8B across all baseline AI agents, achieving a higher average score of 150.7 (±2.3) compared to 125.3 (±30.7) for Llama-8B. This underscores the importance of utilizing a stronger agent to generate high-quality training data for causal reasoning. Importantly, even when using data from Llama-8B, causal knowledge improves performance relative to the absence of causal guidance, where the average score drops to 110.7 (±12.8) as reported in Appx. Tab. 4. We hypothesize that the performance gain observed when using data from Llama-8B arises from its ability to consult not only the current deterministic action selection but also similar past scenarios through the incorporation of causal knowledge.

### D.8 BENEFITS OF CAUSAL KNOWLEDGE INTEGRATION

We divide our analysis into **micro-level failure**, which examines agent behavior within a single environment, and **macro-level failure**, which compares performance across multiple environments.

**Micro-Level Failure.** We analyzed Llama-8B's behavior at 300 timesteps on the *Cramped Room* layout, where our method showed a significant $+36.1\%$ improvement (from 110.7 to 150.7; Fig. 3). Comparing agents with and without CausalPlan, we focused on two failure modes:

**(1) Physically invalid actions.** Calls to `pickup_onion()` while already holding an object (e.g., `hold_onion1` or `hold_dish1`) are reduced with the use of the causal graph. In contrast, valid calls when the agent's hand is empty (`empty_hand1`) increase. Invalid calls dropped from 14 (41%) to 10 (23%), and valid ones rose from 20 (59%) to 33 (76%).

Table 9: Invalid vs. valid `pickup_onion()` calls under different hand states.

| State → Action | Without Graph | With Graph |
|---|---|---|
| `hold_onion1` or `hold_dish1` → `pickup_onion()` | 14 | 10 |
| `empty_hand1` → `pickup_onion()` | 20 | 33 |

**(2) Poor coordination.** The agent avoids redundant pickups when the pot is nearly full (`pot2`) and the other agent already holds an onion (`hold_onion2`). These cases dropped from 2 to 0, reflecting better awareness and coordination from causal integration.

Table 10: Coordination failures with redundant `pickup_onion()` calls.

| State → Action | Without Graph | With Graph |
|---|---|---|
| `hold_onion2, pot_2` → `pickup_onion()` | 2 | 0 |

**Macro-Level Failure.** Across environments, CausalPlan shows the largest improvements on *Cramped Room* (+20.8%), *Asymmetric Advantages* (+18.8%), and *Coordination Ring* (+19.1%), where causal failures such as role confusion, blocking, or redundant actions are common.

In contrast, *Forced Coordination* (+4.9%) emphasizes tight, time-dependent synchronization between agents (e.g., placing too many onions that block the counter while the pot is already full), leaving less room for improvement under our current setup. Notably, we have not yet modeled counter state in the causal graph; incorporating this information could further enhance performance in such layouts.

### D.9 HEATMAP OF LEARNED CAUSAL MATRIX $\mathcal{M}$ ANALYSIS

In Fig. 10 and Fig. 11, we present the causal matrices $\mathcal{M}$ derived from data collected by MEP and Llama-8B, respectively. The inference results using these matrices are detailed in Appx. D.7. To obtain each matrix, the respective agent interacts with the environment for 200,000 steps to gather data, followed by training the SCA model for 500,000 steps on the collected dataset. While both matrices share similarities in many key edges—for example, from `empty_hand1` to `pickup_onion` (edge weights of 0.9 for MEP and 0.8 for Llama-8B) and from `pot_finished` to `fill_dish_with_soup` (0.9 for MEP and 0.8 for Llama-8B) (see Appx. D.12.2 for feature descriptions)—there are important differences that likely contribute to performance variations. For instance, the edge from `pickup_onion` to `put_onion_in_pot` has a weight of 0.6 when using MEP-collected data but is absent (weight 0) with Llama-collected data. Similarly, the transition from `deliver_soup` to `pickup_onion` appears with a weight of 0.7 in the MEP matrix but is missing in the Llama-8B matrix. These differences highlight how the choice of data collection agent influences the learned causal structure, which in turn can impact the effectiveness of downstream inference and control.

Additionally, one may observe that both heatmaps contain several edges that are difficult to interpret, especially those originating from the state of the other agent toward the current action. These edges may carry meaning for the agent but appear unintelligible to humans, or they may be irrelevant. However, these unexpected edges have minimal impact on the inference process, provided the LLM agent does not sample the corresponding actions, thereby eliminating the need to re-calculate the final associated sampling probabilities. This highlights the importance of the general knowledge embedded within the LLM agent, which helps partially eliminate irrelevant edges and leaves only those ambiguities that require causal reasoning. We hypothesize that more advanced causal discovery techniques could further improve the quality of the learned causal graphs by eliminating spurious edges. A simpler alternative might involve hyperparameter tuning of a threshold, where edges with

probabilities below this threshold are removed entirely, or collecting more data. We leave these explorations for future work.

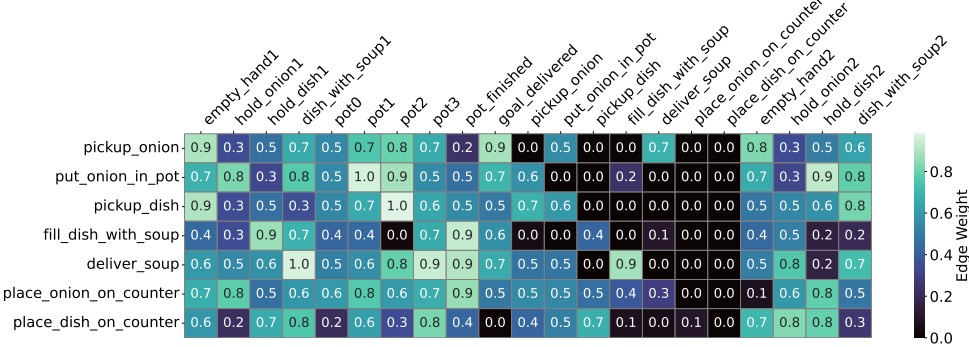

Figure 10: Heatmap of causal graph edge weights obtained from data collected using MEP in the CR layout. The plot illustrates the influence of state features (x-axis) on agent actions (y-axis).

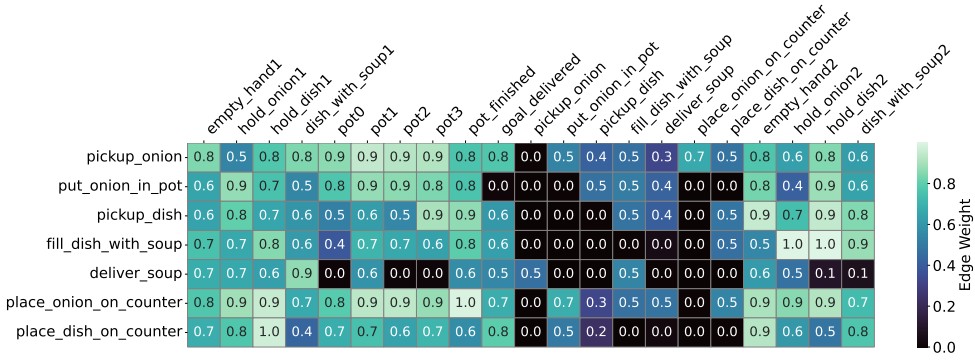

Figure 11: Heatmap of causal graph edge weights obtained from data collected using the Llama-8B backbone in the CR layout. The plot illustrates the influence of state features (x-axis) on agent actions (y-axis).

## D.10 TIME EFFICIENCY ANALYSIS

Learning the causal graph—such as in the CR environment, which involves 21 parent nodes and 7 child nodes—requires approximately 3 hours of training. However, this is a one-time offline process that can be reused across all backbone models, making its cost negligible in the overall training pipeline.

The actual runtime during planning varies depending on the backbone model used. Using NVIDIA h100 GPUs (details in Appx. D.1, we observe the following runtimes for 400 timesteps:

- Gemma-7B and Llama-8B: Approximately 5 minutes without CausalPlan, and around 15 minutes with CausalPlan.
- Qwen-14B: Roughly 16 minutes without CausalPlan, and 41 minutes with CausalPlan.
- Llama-70B: About 40 minutes without CausalPlan, and approximately 68 minutes with CausalPlan.

These results highlight the additional computational cost introduced by causal reasoning. However, the overhead remains reasonable given the observed improvements in policy quality.

## D.11 DETAILED ANALYSIS OF NON-CAUSAL CONDITIONAL MODEL BASELINES

In this section, we provide our hypotheses explaining the results in Tab. 3. Our first hypothesis is that a non-causal supervised model simply reproduces the empirical behavior policy: it imitates demonstrations and therefore cannot meaningfully exceed the demonstrator's performance, which is precisely what we aim to avoid. Such a model lacks any mechanism to depart from demonstrated trajectories, as it has no causal constraints, sparsity, or structural priors specifying which variables influence different components of the action.

We further hypothesize that this baseline performs poorly when the behavior policy $\pi_\beta$ is weaker than the backbone model, since it anchors the backbone to the same suboptimal decisions. To test this, we trained the conditional model $P(a_t \mid s_t, a_{t-1})$ using demonstrations from MEP on two layouts (AA-P1 and CC-P1) where MEP underperforms. As shown in Tab. 3, injecting this non-causal model either degrades performance or provides only limited improvements.

### D.12 HYPERPARAMETERS

#### D.12.1 CAUSALITY AND LLMS HYPERPARAMETERS

**SCA Model**.

Refer to Table. 11.

Table 11: Hyperparameters related to SCA Model

| Parameter | Value | Description |
|---|---|---|
| $N$ | 200,000 | Timesteps used to collect data for buffer $B$. |
| $T$ | 400 | Horizon of each episode. |
| $f_i$ network architecture | MLP | Four hidden layers with dimensions 64, 256, 256, 64; ReLU activations; sigmoid output. |
| Optimizer | Adam | Optimization algorithm used to train $\delta$ and $\eta$. |
| Learning rate | 3e-4 | Step size for gradient updates for $\delta$ and $\eta$. |
| Regularization $\lambda$ | 1e-7 | Regularization strength for parameter estimation. |
| Iterations | 500,000 | Number of training iterations for $\delta$ and $\eta$. |

**LLMs Agent (Build on top of ProAgent framework (Zhang et al., 2024a)**.

Refer to Table. 12.

Table 12: Hyperparameters related to LLMs

| Parameter | Value | Description |
|---|---|---|
| Model sizes | Gemma-7B, Qwen-14B, Llama-8B, Llama-70B | Language model sizes and architectures |
| Temperature | 1.0 | Controls randomness; higher values encourage diverse samples |
| Max new tokens | 256 | Maximum number of generated tokens per output |
| Top-k sampling | 50 | Number of top tokens considered in sampling |
| Top-p sampling | 0.9 | Nucleus sampling threshold (alternative to top-k) |
| Sampling method | Enabled | Sampling is enabled (do_sample=true) |
| retrival_method | recent_k | Parameter of ProAgent framework to retrieve recent history dialogue |
| K | 1 | Parameter of ProAgent framework, the number of history dialogue (default value is 0 or 1) |

**$\gamma$ value in Eq. 4 for each layouts**

Refer to Table. 13.

Table 13: $\gamma$ **value in Eq. 4 for each layout and language model**

| Layout | Gemma-7B | Qwen-14B | Llama-8B | Llama-70B |
|--------|----------|----------|----------|-----------|
| CR     | 0.5      | 0.5      | 0.5      | 0.5       |
| AA     | 0.5      | 0.5      | 0.5      | 0.5       |
| COR    | 0.5      | 0.5      | 0.5      | 0.5       |
| FC     | 0.6      | 0.7      | 0.4      | 0.5       |
| CC     | 0.5      | 0.5      | 0.5      | 0.5       |

### D.12.2  STATES AND ACTIONS FACTORIZATION IN EACH ENVIRONMENT

States and actions factorization used in CR layouts are available in Tab. 14 and for other other layouts are included in Tab. 15.

Table 14: Factorized States and Actions for CR Layout with Descriptions

| Feature | Description |
|---------|-------------|
| empty_hand1 | Controlling agent is not holding any object |
| hold_onion1 | Controlling agent is holding an onion |
| hold_dish1 | Controlling agent is holding an empty dish |
| dish_with_soup1 | Controlling agent is holding a dish filled with soup |
| pot0 | Pot contains 0 onions (empty) |
| pot1 | Pot contains 1 onion |
| pot2 | Pot contains 2 onions |
| pot3 | Pot contains 3 onions (ready to cook) |
| pot_finished | Pot has finished cooking and soup is ready |
| goal_delivered | A soup has been successfully delivered to the goal |
| pickup_onion | Action: controlling agent picks up an onion |
| put_onion_in_pot | Action: controlling agent places an onion into a pot |
| pickup_dish | Action: controlling agent picks up an empty dish |
| fill_dish_with_soup | Action: controlling agent fills a dish with soup from a finished pot |
| deliver_soup | Action: controlling agent delivers a soup to the goal |
| place_onion_on_counter | Action: controlling agent places an onion on the counter |
| place_dish_on_counter | Action: controlling agent places a dish on the counter |
| empty_hand2 | Other agent is not holding any object |
| hold_onion2 | Other agent is holding an onion |
| hold_dish2 | Other agent is holding an empty dish |
| dish_with_soup2 | Other agent is holding a dish filled with soup |

## E  ADAPT TO LONG-HORIZON PLANNING

To adapt CausalPlan to Crafter, a single-agent environment that is often used to evaluate causality-driven methods, we construct the causal matrix $\mathcal{M}$ using only the agent's state and action information. We continue to apply our two-phase causal reasoning framework to guide planning and action selection. In this experiment, we employ Llama-7B as the backbone LLM, and use an underlying PPO policy as our $\pi_\beta$ (similar to Chen et al. (2025)) to collect trajectories and build the causal matrix.

Fig. 12 presents the success rates of CausalPlan (Ours) against Dreamer-V2 (Hafner et al., 2020) and Causal-aware LLMs (Chen et al., 2025) across 22 Crafter tasks. Causal-aware LLMs represent the state-of-the-art approach that integrates causal reasoning into LLM agent planning through prompting. Our method consistently outperforms both baselines, often by substantial margins. In the particularly challenging tasks of *make stone pickaxe* and *make stone sword*, CausalPlan achieves success rates of 5.2% and 6.7%, compared to only 1.3% and 1.6% with Causal-aware LLMs. Likewise,

Table 15: Factorized states and actions for other layouts and their descriptions

| Feature | Description |
|---------|-------------|
| empty_hand1 | Controlling agent is not holding any object |
| hold_onion1 | Controlling agent is holding an onion |
| hold_dish1 | Controlling agent is holding an empty dish |
| dish_with_soup1 | Controlling agent is holding a dish filled with soup |
| pot0_0 | Pot 0 contains 0 onions (empty) |
| pot1_0 | Pot 0 contains 1 onion |
| pot2_0 | Pot 0 contains 2 onions |
| pot3_0 | Pot 0 contains 3 onions (ready to cook) |
| pot_finished_0 | Pot 0 has finished cooking and soup is ready |
| pot0_1 | Pot 1 contains 0 onions (empty) |
| pot1_1 | Pot 1 contains 1 onion |
| pot2_1 | Pot 1 contains 2 onions |
| pot3_1 | Pot 1 contains 3 onions (ready to cook) |
| pot_finished_1 | Pot 1 has finished cooking and soup is ready |
| goal_delivered | A soup has been successfully delivered to the goal |
| pickup_onion | Action: controlling agent picks up an onion |
| put_onion_in_pot | Action: controlling agent places an onion into a pot |
| pickup_dish | Action: controlling agent picks up an empty dish |
| fill_dish_with_soup | Action: controlling agent fills a dish with soup from a finished pot |
| deliver_soup | Action: controlling agent delivers a soup to the goal |
| place_onion_on_counter | Action: controlling agent places an onion on the counter |
| place_dish_on_counter | Action: controlling agent places a dish on the counter |
| empty_hand2 | Other agent is not holding any object |
| hold_onion2 | Other agent is holding an onion |
| hold_dish2 | Other agent is holding an empty dish |
| dish_with_soup2 | Other agent is holding a dish filled with soup |

in *make iron pickaxe* and *make iron sword*, CausalPlan succeeds where both Causal-aware LLMs and Dreamer-V2 fail. These improvements in individual tasks are reflected in the aggregated final score (Tab. 16), where CausalPlan achieves a higher final score of 16.7%, surpassing Causal-aware LLM (14.6%) and Dreamer-V2 (10.0%). These results highlight that prompting alone is insufficient for robust long-horizon planning, whereas our method provides more reliable improvements by grounding decisions in causal structure.

Table 16: Scores (mean ± std) of our method and baselines on 22 Crafter tasks.

| Method | Final Score (%) |
|--------|-----------------|
| Dreamer-V2 | $10.0 \pm 1.2$ |
| Causal-aware LLMs @ 1M | $14.6 \pm 2.2$ |
| **CausalPlan (Ours) @ 1M** | $\mathbf{16.7 \pm 1.2}$ |

## E.1 VISUALIZATION OF CAUSAL GRAPH

To evaluate whether our method recovers meaningful structure, we validate it against the known recipe (domain knowledge) and physical constraints defined by the Overcooked-AI rules. We use the Cramped Room (CR) layout for illustration because it contains the smallest set of actions/features, making the learned graph easy to interpret.

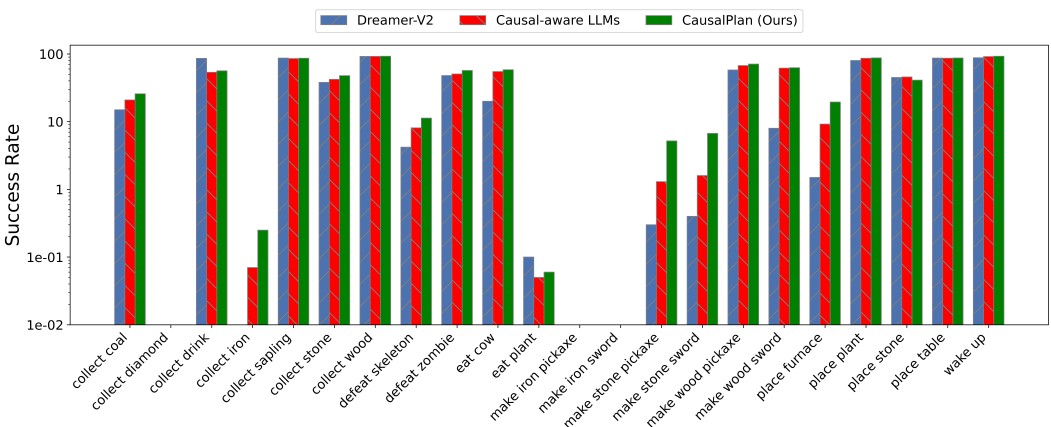

Figure 12: Success rates of obtaining 22 achievements in log scale @1M steps.

### E.1.1 DOMAIN KNOWLEDGE VERITIFCATION

To construct the recipe graph, we begin from the soft causal action matrix (learned under the MEP behavior policy; Fig. 10). We then remove any bidirectional edges by keeping only the direction with the higher learned weight, yielding a directed acyclic graph (penalize length-2 cycles). The structure (see Fig. 13) correctly identifies the canonical recipe sequence required by the environment: first pick up the onion, then put the onion in the pot, next pick up the dish, fill the dish with soup, deliver the soup, and finally return to pick up another onion. These edges (highlighted in red in the figure) match the ground-truth procedural dependencies of the cooking task.

The other edges (black) show that the matrix $\mathcal{M}$ does not directly correspond to the MEP policy, but is a plausible indirect dependency arising from the structure of the recipe. For example, while a trained model like MEP does not execute pick up the onion to pick up the dish consecutively, such a dependency is reasonable because a dish is typically picked up only after onions have already been placed in the pot, and the learned causal graph manage to capture this broader sequencing.

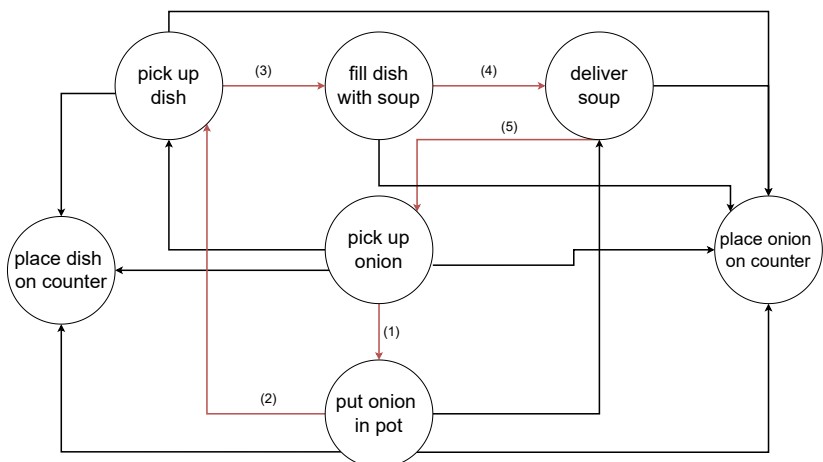

Figure 13: Directed recipe graph constructed from the soft causal action matrix. Red edges indicate the canonical procedural sequence for the cooking task, correctly reflecting the environment's required steps, while black edges represent plausible indirect dependencies captured by the learned causal structure. The graph demonstrates that the model identifies both direct and broader sequencing relationships in the recipe.

### E.1.2 PHYSICAL CONSTRAINT VERITIFCATION

To illustrate physical constraint, we look at local causality. We specifically look at action to pick up the onion (see Fig. 14). Applying a hard intervention with a threshold of 0.7 (keeping only edges with scores above the threshold) identifies the relevant state features that causally influence this action: empty_hand1 and empty_hand2. While some additional edges are present, they do not contradict task logic (for instance, after the pot already contains two onions, the agent is still required to pick up more onions to complete the recipe) and may reflect broader dependencies; importantly, their inclusion is associated with improved task performance. This demonstrates that even at the level of individual actions, the learned causal action matrix $\mathcal{M}$ captures meaningful dependencies between state features and next actions, supporting the interpretability and correctness of the inferred graph.

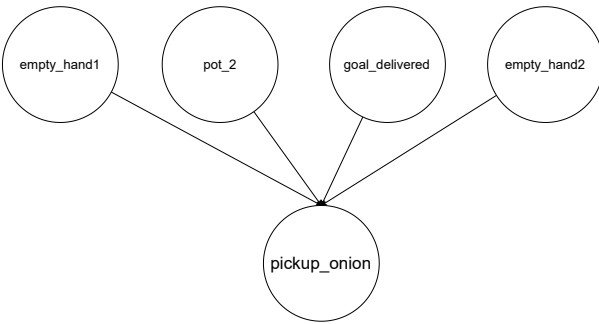

Figure 14: Local causal dependencies for the action pick up onion, derived from the learned causal action matrix with a hard intervention threshold of 0.7. The highlighted edges show that the action is causally influenced by the state features empty_hand1 and empty_hand2, reflecting the physical constraint of the environment that the agent can only pick up an onion when a hand is free.

## F DISCUSSION OF BROADER IMPACTS

This work represents an important foundational step toward integrating causal reasoning into multi-agent planning with large language models (LLMs). Our causality-driven framework aims to improve the safety, efficiency, and interpretability of collaborative AI systems by enabling agents to better understand the consequences of their states and actions. Although primarily exploratory and not yet intended for real-world deployment, the results demonstrate promising potential for advancing multi-agent coordination.

At this stage, we do not expect any direct negative societal impacts, as the framework requires further development and validation before practical use. Nevertheless, as autonomous multi-agent systems mature, concerns related to fairness, reliability, misuse, and broader ethical implications will become increasingly important. Addressing these challenges through responsible design, transparency, and rigorous evaluation will be critical to ensure the safe and trustworthy deployment of such systems in the future.

