# OpenReview forum: "CausalPlan: Empowering Efficient LLM Multi-Agent Collaboration Through Causality-Driven Planning"
_ICLR.cc/2026/Conference — Submitted to ICLR 2026_

### Official Review · Reviewer_AtpH · 2025-10-27

**Soundness:** 2
**Presentation:** 3
**Contribution:** 3
**Rating:** 4
**Confidence:** 2

**Summary:**

The paper introduces CausalPlan, a planner for two-agent collaborative tasks that reduces invalid actions by learning and exploiting a sparse policy structure over decisions. Given a set of trajectories, each timestep is factorized into binary features (agent states and environment state) and a one-hot previous action of the agent being controlled. Then a per-action head is trained with a sparsity mask using NLL (alternating masks/weights), indicating how much each input feature is used when choosing that action. The masks form a matrix whose entries reveal the propensity of choosing an input feature when choosing a decision (action). At test time, the agent a) prompts an LLM to propose candidate high-level actions, b) the actions are pruned based on feasibility using external rules/grounding, and c) each remaining candidate is scored by summing the mask weights from the currently active features. The score is blended with LLM scores  via a convex combination. If there are no valid actions, a fallback regime is used where the chosen action is the top-scoring action from the learned matrix. Empirically, this cuts invalid actions and improves cooperative rewards across layouts and LLM backbones.

**Strengths:**

- Tackles a real failure mode of LLM agents in collaborative tasks
- Fills a real gap between pure prompting and heavy world-modelling.
- Simple training pipeline: Learns from trajectories using a standard NLL+sparsity objective
- The policy-structure matrix gives insight into which inputs support which actions.
- The method shows consistent empirical gains over strong baselines.
- The system is partner-aware (by including the partner’s state in the feature set) without requiring access to or having to learn the partner’s policy
- Inference is simply masked feature sum+convex blend of LLM action probabilities, far cheaper than replanning/multiple model rollouts.
- Deployability is practical, given that the same assumptions hold. There is minimal friction into existing agent stacks given the learned matrix; one matrix lookup, one re-weighing step.
- The appendix is comprehensive and code was provided, facilitating reproducibility.

**Weaknesses:**

I will combine weaknesses, remarks, and questions in one section for readability.

The comments below reflect my current reading of the paper and appendix; if I’ve misread any definitions or misinterpreted any claims, I  welcome pointers and will happily revise.

To my understanding, the paper does not build a causal model of the world, though the writing sometimes suggests it is. The learned object is a policy-structure over observed features that predicts the next action, not a dynamics model one could query with do operators or counterfactuals. The SCA takes parents $(a_{t-1}, s_t)$. This models cause-effect relationships within the agent’s decision process, not the environment’s physics or tasks dynamics

According to the definition at L229-L232, each row of $M$ corresponds to a possible next action and each column to a state of past-action features. Each entry is the learned probability that feature $j$ influences action $i$, given the learned structure. Querying $M$ sums the active parent entries to produce a "causal score". In effect, from my understanding, higher sum implies that the model has learned that the currently active features are predictive parents of that next action. This is not a causal effect estimate in the Pearl sense.

The proof’s conclusion (L812-L814) that "the causal action matrix … faithfully reflect the true cause-effect relations among states and actions" reads too strongly. What is captured is a sparse dependence over $(s_t, a_{t-1})\to a_t$, which is a property of the actor’s policy, not of the world’s causal structure.

A (hard) intervention (if atempted) would possibly be toggling columns $j$ (making a feature active/inactive) and seeing how the score changes. The paper does not define or use interventional queries over an SCM over the environment’s variables. Concretely, the method learns a decision structure, not environment causality.

Can you clarify the above in the paper?

Minor: “intervention” is used informally (L277, re-prompting) which can be confusing in a section that discusses causal effects and structural causal action models.

Furthermore, there are two distinct issues with the proof:

A) Proof-method mismatches
- Estimator mismatch; The appendix analyses ridge on fixed basis features and reads parents from the support of $W$. The method itself trains neural Bernoulli heads via NLL and learns $\eta$ jointly. These are not equivalent and, as far as I can tell, one does not imply the other.
- ANM vs classifier; The proof invokes ANM-style identifiability but the trained model is a binary classifier
- Acyclicity gap: The proof assumes a DAG. The method’s heuristic "zero the smaller of each bidirectional pair" only removes 2-cycles, leaving (3+)-cycles in the action-action portion of the graph). Take for example the following relations: $W(a \to b) = 0.5 > 0.3 = W(b \to a)$, $W(b \to c) ‎ = 0.5 > 0.3 = W(c \to b)$, $W(c \to a) = 0.5 > 0.3 = W(a \to c)$. The heuristic would remove the three arrows $b \to a$, $c \to b$, $a \to c$ but $a \to b \to c \to a$ remains. This is inconsequential when the learned matrix is used in a feedforward manner as done in the paper, but the claim (L236-238)"…ensure DAG property of a standard SCM…" is not correct.
- Observability: The proposition (L220) states $a_t$ is unobservable, but the proof in the appendix uses an observed $a_t$ to train the SCA model. Clarifying unobserved at test time, observed during training would help.

Any of the above, in my view, render the statement proven in the appendix inapplicable to the method in the main paper (apart from the observability claim but that’s a wording inconsistency).

Questions: Either restate and prove a proposition for the actual model class and estimators, align the method to the ridge/basis estimator in the appendix or clearly state that this is a motivating surrogate that does not apply to the method.


B) Standalone proof issues

When considering the proof itself in isolation:

- The specific regularizing conditions/assumptions are not clearly stated (i.e., L728: "identifiability of causal direction relies on the function class having sufficient expressiveness and satisfying certain regularity conditions (e.g., nonlinearity, invertibility)". Invertibility, as far as I can tell, is not relevant to the proof. Please state the exact assumptions used.
- Ridge regression is used and the parents are identified by the support of $W_i$ (L808) with the claim (L809): "one can recover the graph structure by examining which entries of $W_i$ are significantly nonzero, using thresholding or statistical tests.". L2 regularisation has no sparsity guarantees and typically yields dense solutions. How exactly is this step justified and implemented?
- The proof assumes causal faithfulness but the collaborator’s actions are not modelled (L899-901). If $u$ denotes the collaborator’s action, an implicit assumption is made: $a_t \perp u_{t-1} \mid (s_t, a_{t-1})$. If $s_t$ is intended to be sufficient to mediate all effects of the parent’s last action, clearly state so, otherwise sufficiency is violated.
- Eq (8) instantiates the dataset as sequences of the form $(s_t, a_{t-1}, a_t)$. Clearly state what is observable and what is not.

Furthermore:
- L244-246: Can you clarify how the actions are sampled by the LLM and how they are scored?
- In the action pruning step, the method assumes access to an external verifier of feasibility. What does this verifier look like? What happens when such a verifier is unavailable (i.e., real-world robotics)? If it is required, can you add this as an explicit assumption?
- Why does CausalPlan underperform in some settings (Table 1)? A rief discussion of failure modes would help.
- L318: A short description of the baselines in the main text (with details in the appendix) would make the experiments easier to follow.
- What is the impact of removing the previous action feature/removing partner-state features or adding the partner’s previous action?
- How well does an SCA trained on trajectories from a behaviour policy paired with a partner transfer when deployed with different partners?

**Questions:**

See weaknesses

---

> ### Author Response · Authors · 2025-11-19
> **Reply Reviewer AtpH (1/3)**
>
> We sincerely thank the reviewer for the thoughtful and detailed feedback regarding the proof, terminology, and the causal interpretation of CausalPlan. Your comments have helped us reflect carefully on the clarity and rigor of our presentation. We address the main points below and provide additional evidence and discussion where appropriate. We hope that these clarifications will improve the understanding of our work, and we greatly appreciate the opportunity to engage further with your insights to strengthen the manuscript.
>
> # W1: Policy-Level Causal Interpretation
>
> Our model captures **cause-effect relationships within the agent’s decision process**, not the environment’s physics or task dynamics. To avoid any misunderstanding, we have rephrased the conclusion of the identifiability proof (lines 867–869 of the revised manuscript) as follows:
>
> **"As a result, the Structural Causal Action model learned by CausalPlan captures a sparse pattern of causal dependencies at the policy-level, providing a stable and interpretable approximation of the underlying decision-making process."**
>
> While the model does not explicitly encode environment dynamics, some aspects of the environment are still reflected in the SCA structure. For example:
> - The recipe the agent follows to complete the task is reflected in the learned dependencies (see revised paper Appendix E.1).
> - The edge from *empty_hand* to *pickup_onion* captures a physical constraint: the agent can only pick up an onion when at least one hand is free.
>
> These examples illustrate that the learned policy-level causal structure aligns with intuitive task logic, even without modeling the underlying environment.
>
> # W2: Conceptual Nature of the Identifiability Proof
>
> We acknowledge that the identifiability proof in the appendix is **not a formal guarantee for the exact SCA model used in CausalPlan**. Instead, it serves as a **conceptual or surrogate illustration** to provide intuition for why the policy-level causal structure can, in principle, be recovered under standard identifiability conditions.
>
> Key points:
> - The proof analyzes a simplified setting (ridge regression over fixed basis features), which differs from the neural Bernoulli heads used in practice.
> - Certain assumptions in the proof (e.g., ANM-style identifiability) do not strictly hold for the actual model class.
> - The proof demonstrates the feasibility of recovering sparse causal dependencies conceptually, rather than establishing formal guarantees for the trained neural model.
>
> We have revised the manuscript to clearly present the appendix proof as a simplified, motivating illustration, and explicitly clarify where its assumptions and conclusions differ from the practical CausalPlan model. **Additionally, we have added Appendix B: Conceptual Comparison of Ridge Regression and Bernoulli NLL, which connects the ridge regression analysis in the proof to the actual neural Bernoulli heads used in CausalPlan**. This appendix shows that for small prediction errors, the NLL gradient approximates the ridge regression gradient, thereby contextualizing the theoretical discussion even in the absence of formal guarantees for the trained neural model.
>
> # W3: Clarification on Intervention Mechanism
>
> While classical hard interventions in an SCM involve manipulating environment variables, CausalPlan performs a **policy-level intervention**: during training, functional parameters $\delta$ are learned under edge configurations sampled from a Bernoulli distribution based on the current graph belief: ~Ber(sigmoid($\eta$)). Each sampled configuration effectively toggles which parent features are active for that mini-batch, analogous to a hard intervention at the policy level. This procedure trains the model under many plausible structural configurations, yielding robust dependencies between state-action features and next actions.
>
> This Bernoulli sampling mechanism was missing in the original appendix and has now been added (lines 951–961).
>
> ---

---

> ### Author Response · Authors · 2025-11-19
> **Reply Reviewer AtpH (2/3)**
>
> # W4: Sufficiency and Partner's Actions
>
> - We acknowledge that the proof assumes causal faithfulness while the partner’s actions are not explicitly modeled. In principle, this violates the assumption of causal sufficiency, since partner action could mediate dependencies not captured by the observed state.
>
> - However, when we implement CausalPlan without explicitly modeling the partner’s actions, this version empirically yields better performance than including the partner’s actions. It also outperforms versions that remove either the previous action feature or partner-state features. We hypothesize that conditioning on the agent’s observation $s_t$ is often sufficient to capture the relevant effects of decision-making in our task, even if strict causal sufficiency is technically violated.
>
> - That said, we still need to include the previous action of the current controlled agent to correctly capture temporal dependencies and enforce causal consistency in the decision process, ensuring that the model can reason about action sequences and not just instantaneous state effects.
>
>
> # W5: L2 Regularization and Sparsity
>
> - We agree that L2 regularization does not theoretically guarantee sparsity, as it typically yields dense solutions. However, in practice, it is common in prior work to **combine L2 with a pre-defined threshold** on the coefficients to enforce sparsity in the recovered graph structure.  Specifically, after fitting the model, coefficients whose absolute values fall below the chosen threshold are set to zero, effectively identifying the parents of each node.
>
> # W6: Terminology Clarification and Partial DAG
>
> - We agree that the term “intervention” (line 277) can be misinterpreted as a causal operator. To avoid ambiguity, we now refer to this as a **“fallback strategy”** (line 285 of the revised manuscript).
>
> - In addition, we have updated the manuscript (line 245-247) to clarify and correct previous statements that implied full DAG guarantees. Specifically, we note that: **To enforce a partial DAG structure in the causal graph, we compare the coefficients for each pair of mutually connected nodes in $\mathcal{M}$ and set the smaller coefficient to zero, which removes 2-cycles."**
>
> # Q1: LLM Action Sampling
> In our setup, each action is represented as a **text token** (or a short sequence of tokens) generated by the LLM. Actions are sampled from the predicted next-token distribution using a **temperature parameter** $T$: $P(a_i) = \frac{\exp(z_i / T)}{\sum_j \exp(z_j / T)},$ where $z_i$ are the logits over the token vocabulary. We typically use a **higher temperature** during sampling to encourage exploration of diverse action candidates.
>
> Each action (token or token sequence) is scored using the log-probability assigned by the LLM. In our prompt, we constrain the LLM to output exactly the action from a predefined list of valid actions. However, if the LLM fails to produce a valid action, we follow the causal backup plan: *"pickup_onion"* sampled according to the softmax over the token vocabulary, scored by  log P(pickup\_onion | $s_t$), where $s_t$ represents the current context window including observations and past actions.
>
> # Q2: Action feasibility verifier
>
> In the action pruning step, we assume access to an external verifier that checks whether a candidate action is feasible in the current state. This verifier is often available in simulation or structured environments, as commonly assumed in prior work. In real-world robotics, similar feasibility checks can often be implemented using perception modules (e.g., depth sensors, collision detection) or constraints in motion planning.  We have added this explicitly as an assumption in the revised manuscript line 256
>
> # Q3: Failure mode
>
> We hypothesize that, even though CausalPlan captures policy-level causal structure, it still relies on the LLM's internal linguistic reasoning. In cases where the LLM assigns overly deterministic probabilities to certain actions, the causal reasoning provides limited benefit, as the action selection is dominated by the LLM's pre-existing biases rather than the learned causal dependencies.
>
> # Q4: Impacts of feature ablation
>
> As mentioned previously in our answer to W4, removing the previous action feature of the agent or the partner’s state features from the input consistently degrades performance, indicating that both sets of features are important to capture dependencies relevant to causal reasoning at the policy-level.
>
> # Description of baseline in main text
> We agree with the reviewer’s suggestion. Providing a short description of the baselines in the main text, while keeping full details in the appendix, would indeed make the experiments easier to follow. We will consider re-arranging this section after the discussion phase to improve clarity and readability.

---

> > ### Author Response · Authors · 2025-11-19
> > **Reply Reviewer AtpH (3/3)**
> >
> > # Q5: Transfer to new partners
> >
> > We appreciate the reviewer’s question, which highlights a point that may not have been clearly explained. In our evaluation, we measure how well an SCA trained on trajectories from a behaviour policy paired with a specific partner transfers when deployed with different partners. Concretely, we train CausalPlan using trajectories collected between 2 MEP agents and then deploy the LLM with matrix $\mathcal{M}$ with all other agents (SP, COLE, MEP and FCP), reporting the average performance across these deployments. We mention this in line 316-318 of the revised manuscript. This setup allows us to assess the robustness of the learned policy-level causal structure to variations in partner behaviour.

---

### Official Review · Reviewer_RXj8 · 2025-11-01

**Soundness:** 2
**Presentation:** 4
**Contribution:** 2
**Rating:** 4
**Confidence:** 4

**Summary:**

This paper proposes CausalPlan, a framework that purports to integrate explicit structural causal reasoning into LLM-based multi-agent planning. The core contribution is a Structural Causal Action (SCA) model that learns relationships between prior actions, current states, and future actions from agent trajectories. These learned relationships are encoded in a ``Causal Action Matrix'' $M$, which is then used to reweight LLM-generated action probabilities during planning. The authors evaluate their approach on the Overcooked-AI benchmark across multiple LLM backbones and show empirical improvements in task success rates and reductions in invalid actions.

**Strengths:**

1. Consistent gains across very different LLM backbones and layouts, not just one setup.

2. Human-partner results are stronger than baselines and include statistical testing; several settings reach p<0.05 and none show degradation when the method is enabled.

3. The causal backup plan is an effective recovery mechanism when the planner proposes no valid actions; ablation shows it adds measurable benefit beyond the two-prompt tweak.

4. The framework exposes a causal action matrix and publishes heatmaps, giving a degree of interpretability about which state/action factors influence next actions.

5. Robustness to who collects the data for the buffer; using a stronger behavior policy helps, but even weaker LLM-collected data still benefits from the causal integration.

6. Sensitivity analysis of the γ weighting shows a broad sweet spot.

7. Explicit DAG enforcement by zeroing the weaker direction in any bidirectional pair prevents cycles in the learned structure.

8. Modular drop-in over ProAgent with open-source LLMs, keeping the rest of the stack intact and making replication or extension straightforward.

9. Extends beyond Overcooked to a long-horizon single-agent benchmark (Crafter) and outperforms both a causal-prompting baseline and Dreamer-V2 at 1M steps.

12. Prompting design separates analysis from action selection, making the action extraction unambiguous; ablation indicates the components introduced to capture causal relationships drive most of the lift.

**Weaknesses:**

1. The framework learns from trajectories generated by a fixed behavior policy in Overcooked-AI, which means each action is conditioned on the policy’s internal decision process. Since actions aren’t randomized or independently manipulated, the data are observational, not interventional.

2. The Structural Causal Action model optimizes a likelihood loss ( -\log P(a_t \mid s_t, a_{t-1}) ), which captures conditional correlations rather than causal effects ( P(a_t \mid s_t, \text{do}(a_{t-1})) ). Without interventions or counterfactual adjustments, the learned structure reflects co-occurrence patterns, not causal mechanisms.

3. Although Overcooked-AI’s environment is deterministic, the data collection process is not interventionally controlled. The simulator ensures that actions deterministically affect states, but the trajectories used for learning are policy-dependent rollouts, not samples from systematically applied interventions.

4. Because the same policy governs both state visitation and action choice, correlations between (s_t) and (a_t) can arise from shared dependencies on unobserved latent factors such as internal LLM reasoning or high-level strategy. The model treats these as causal links.

5. The binary feature encoding used for states and actions is a coarse abstraction of the full simulator state. Hidden variables like spatial positioning or timing can confound state–action dependencies, violating causal sufficiency.

6. The framework’s only verification is improved prediction accuracy and task performance, which measure behavioral alignment, not causal correctness. A model can be highly predictive while causally wrong.

7. The paper’s theoretical identifiability proof relies on assumptions such as additive noise, faithfulness, full observability, and acyclicity, none of which are verified in Overcooked-AI. There is no empirical evidence that these assumptions hold in practice.

8. Each entry of the causal action matrix represents a learned dependency weight, not an intervention-derived causal coefficient. The matrix is effectively a correlation matrix with sparsity regularization.

9. The observed reduction in invalid actions and improved cooperation may result from regularized prediction smoothing or bias correction, not genuine causal reasoning. The gains demonstrate utility, not causal validity.

10. Because the learned structure reflects policy-specific correlations, the matrix may not transfer to different partners, environments, or task variations, contradicting the stated goal of causal generalization.

**Questions:**

1. How do you distinguish causal effects from correlations when all data come from a fixed behavior policy π₍ᵦ₎? What is your formal definition of causation in this context?

2. Can you provide empirical evidence that the faithfulness, causal sufficiency, and additive noise assumptions hold in Overcooked-AI? For instance, conditional independence tests, checks for unobserved confounders, or validation of the additive noise model?

3. What would an intervention experiment look like to validate your learned causal structure? For example, could you force an agent to take actions inconsistent with M and measure the deviation in outcomes?

4. Why not compare against a model that learns P(aₜ | sₜ, aₜ₋₁) with standard neural networks (e.g., feedforward or recurrent) without causal constraints? Does the DAG structure and sparsity actually matter, or are the gains from additional learned features?

5. How does performance degrade when the partner policy changes? Does your “causal” matrix M transfer to new partners, or is it partner-specific?

6. Can you show that the learned dependencies correspond to true causal mechanisms rather than artifacts of π₍ᵦ₎? For instance, by comparing M learned from different behavior policies?

7. Have you tested whether M changes systematically under distributional shift? This would be evidence of instability inconsistent with causal invariance.

8. Why is binary feature encoding sufficient when it discards causally relevant information such as spatial distances, timing, and interaction history?

9. What is the causal graph G you claim to identify? Can you draw it explicitly (not just heatmaps of M) and verify it against ground truth or domain knowledge?

10. In Proposition 1, you assume aₜ is “unobservable” during training, but clearly you observe aₜ in the trajectory data 𝓑. Can you clarify this apparent contradiction?

**Details Of Ethics Concerns:**

No ethical concerns.

---

> ### Author Response · Authors · 2025-11-19
> **Reply Reviewer RXj8 (1/4)**
>
> We sincerely thank the reviewer for their careful and detailed reading of our work. We appreciate the recognition of our consistent gains across LLM backbones, the interpretability of the causal action matrix, robustness to behavior policy choice, and thorough ablations. We also value the constructive questions regarding correlation vs. causation, partner generalization, interventions, and feature encoding. We have addressed these in detail in the revised manuscript with clarifications, additional experiments, and expanded discussion. We hope these updates address the reviewer’s concerns and respectfully invite reconsideration of the paper’s assessment given the following clarifications.
>
> # Q1: Clarify the formal definition of causation
> We agree that, strictly speaking, our method does not recover the environment’s causal structure in the Pearl sense. **What we mean in this context is that the causal action matrix $\mathcal{M}$ captures causal influence within the agent’s decision process** (as also pointed out by Reviewer AtpH): it encodes how prior state and action features systematically affect subsequent action choices. **By simulating interventions on prior features (when optimizing $\delta$ and $\eta$), the agent identifies edges that reflect consistent, predictive dependencies.**
>
> - To illustrate, consider a human driver navigating city traffic. The driver’s next action—accelerating or turning—depends on features such as the car’s current speed, traffic light state, or the behavior of nearby vehicles. The agent-level causal influence (analogous to $\mathcal{M}$) captures how these observed features systematically affect the driver’s next action. This does not require knowledge of the full environment-level causal structure. Yet, understanding these systematic dependencies allows the driver to plan actions effectively.
>
> Similarly, in CausalPlan, $\mathcal{M}$ represents how prior state and action features influence the agent’s next actions. While these are not causal effects of environment variables, they legitimately capture the internal causal mechanisms of the agent’s policy.
>
> # Weakness and Q2:  Causal Sufficiency, Faithfulness, Additive Noise.
>
> - The design of Overcooked makes unobserved confounding extremely unlikely. **In Overcooked-AI, all state variables that influence the next legal action are explicitly encoded: grid layout, object positions, agent positions, carried objects, and recipe progress. There are no hidden intentions or stochastic forces**.
>
> - Although deterministic systems often technically violate strict faithfulness, **in practice Overcooked transitions exhibit a stable mapping between factor changes and action choices**. When a relevant factor changes (e.g., “agent is carrying onion”), the agent’s future actions change in a predictable, structured way (e.g., “place-onion” becomes available and likely). If the factor doesn’t change, the available actions remain the same and stable. So, for all practical purposes, the system behaves in a way that is “faithful”: changes in causal factors consistently affect subsequent actions, allowing the SCA model to learn meaningful dependencies.
>
> - In our Structural Causal Action (SCA) model, the noise term does not come from the environment. Instead, it **captures all stochasticity coming from the behavior policy that generated the data**. If you use an LLM policy like Llama-8B → the noise term reflects Llama-8B’s variability (LLMs are inherently noisy: token sampling, temperature, prompt sensitivity and their decisions can vary even under identical states). If you use a pretrained MEP policy → the noise term reflects MEP’s variability (MEP may choose different valid actions in similar states and may also have randomness depending on the partnering agent).

---

> ### Author Response · Authors · 2025-11-19
> **Reply Reviewer RXj8 (2/4)**
>
> # Weakness and Q3:  Intervention.
>
> We thank the reviewer for the detailed comments regarding intervention and causal learning. We agree with the reviewer that the data in this case is observational data. **However, we would like to clarify that CausalPlan does perform a form of intervention during the optimization of the two parameters $\delta$ and $\eta$, even though this was not made fully explicit in the original manuscript**. Specifically:
> - In our method, the functional model parameterize $\delta$ encode the effects of state, action → next-action transitions. When we optimize $\delta$ (keeping $\eta$ fixed), we are effectively applying interventions on this functional/generating model, simulating the effect of perturbing prior state or action features on subsequent action features. **Crucially, we do not apply the “correct” graph structure directly as mask. Instead, we draw a number of graph configurations from a Bernoulli distribution based on current graph belief**: ~Ber(sigmoid($\eta$)). Each sampled graph specifies which edges are “on” or “off” between features, and we train $\delta$ under these different causal hypotheses. T
> - In the second phase, we fixed $\delta$ while optimizing $\eta$. **Under the learned $\delta$, we then adjust $\eta$ toward configurations that lead to better predictions by backpropagating the prediction loss**. In other words: edges that improve predictive performance get reinforced, edges that hurt performance get down-weighted. This gradually moves the graph belief toward the structure that best matches the interventional evidence generated in Phase 1.
>
> Although the data we collect are observational as pointed out, the graph learning process itself incorporates interventions. This is why, even without explicit interventional data collection, our method can capture causal influence patterns beyond mere correlations. This two-phase approach for learning structural causal models from observational data with interventional simulation follows the method introduced by [1] and has been used in prior causality integrated research in RL [2,3,4]. We note that the Bernoulli sampling details were missing from the original Appendix, which may have caused confusion. We have updated the Appendix C.1 to explicitly reflect this process.
>
> [1] Ke, N. R., Bilaniuk, O., Goyal, A., Bauer, S., Larochelle, H., Schölkopf, B., ... & Bengio, Y. (2019). Learning neural causal models from unknown interventions. arXiv preprint arXiv:1910.01075.
>
> [2] Hu, X., Zhang, R., Tang, K., Guo, J., Yi, Q., Chen, R., ... & Chen, Y. (2022). Causality-driven hierarchical structure discovery for reinforcement learning. Advances in Neural Information Processing Systems, 35, 20064-20076.
>
> [3] Zhang, Y., Du, Y., Huang, B., Wang, Z., Wang, J., Fang, M., & Pechenizkiy, M. (2023). Interpretable reward redistribution in reinforcement learning: A causal approach. Advances in Neural Information Processing Systems, 36, 20208-20229.
>
> [4] Yu, S., & Lu, C. (2024). Adam: An embodied causal agent in open-world environments. arXiv preprint arXiv:2410.22194.
>
> # Q4: Compare against $P(a_t \mid s_t,a_{t-1})$
>
> We have added a comparison against  $P(a_t \mid s_t,a_{t-1})$ to Section 4.5 of the revised version (line 426-461). **A non-causal supervised model simply reproduces the empirical behavior policy: it imitates the demonstrations and cannot meaningfully exceed the demonstrator’s performance (precisely what we aim to avoid)**. Critically, this approach cannot outperform the behavior policy because it lacks any mechanism to depart from the demonstrated trajectories. It has no causal constraints, no sparsity, and no structural priors guiding. **We hypothesize that such a baseline would not perform well when the behavior policy is weaker than the backbone model, since it will anchor the backbone      model to the same suboptimal decisions**. Here we provide the results of $P(a_t \mid s_t,a_{t-1})$ model trained on MEP demonstrations and combined with Llama-70B by averaging their action-selection probabilities (we call $MEP_{guided}$) and $MEP_{backup}$ replacing our causal backup mechanism with action with probability $P(a_t \mid s_t,a_{t-1})$.
> - Simply used $MEP_{guided}$ degrades the performance of the backbone.
> - Use $MEP_{backup}$ improves the performance of the backbone but not as much as CausalPlan. We hypothesize that this is because $MEP_{backup}$ only handle the cases when back up is needed (ignoring cases when backbone LLM select action that is physically valid but sequentially incorrect).
>
> | Layout | Llama-70B (backbone) |  $MEP_{guided}$ | $MEP_{backup}$ | CausalPlan|
> |----------------|-------------|----------------------|---------------------|---------------------|
> | AA-P1             |  248.0 +- 22.7       | 207.3 +- 19.4   | 257.3 +- 9.2 | 266.7   +- 16.7              |
> | CC-P1           | 89.3 +- 32.3        | 80.3 +- 9.2     | 90.7  +- 12.9         | 112.0   +- 6.9

---

> ### Author Response · Authors · 2025-11-19
> **Reply Reviewer RXj8 (3/4)**
>
> # Q5: Partner Specific
>
> We thank the reviewer for the question regarding partner generalization. **We would like to clarify that the learned causal action matrix $\mathcal{M}$ is not partner-specific**. Our empirical evaluation supports this:
> - When collecting data, we use trajectories from the MEP partner paired with MEP (this is to avoid biasing the causal action matrix toward a specific partner policy).
> - **During evaluation, the backbone LLMs with CausalPlan applied were tested against all partnering baselines, not just MEP**. The LLMs are unaware of which partner they are paired with, and the results reported in Table 1 reflect the average performance across all partner pairings (COLE, FCP, MEP, PBT, and SP). For the baselines, the average is computed against all other partners excluding themselves and the LLM. For example, for MEP, this average includes pairings with CausalPlan, COLE, FCP, PBT, and SP.
>
> # Q6: True Causal Mechanism
>
> We believe that Matrix $\mathcal{M}$ reflects true causal mechanism rather than just merely capturing action selection probability of the behavior policy for the following reasons:
>
> - Firstly, as shown in Table 5 (Appendix D.4), while MEP underperforms in the AA and FC layouts, CausalPlan (Llama-70B) still achieves strong results. In addition, MEP’s performance with different partners differs from that of CausalPlan. For example, from the table below, we see that MEP performs best when paired with PBT, whereas both Llama-8B and Qwen-14B with matrix $\text{CausalPlan}_{MEP}$ perform best with COLE. Similarly, while MEP performs better with FCP than with SP, the CausalPlan LLMs perform better with SP than with FCP. This indicates that performance is not tied to the specific partner used during data collection.
>
> | Partner | MEP | Llama-8B | Qwen-14B|
> |----------------|-------------|----------------------|----------------|
> | COLE             |  153 +- 23.1      |  253 +- 11.5                 | 253 +-11.5         |
> | FCP            | 153 +- 46.1        |  186 +- 30.5            | 226 +-23.1
> | PBT           | 187 +- 11.5        | 193 +- 23.1               | 240 +- 0          |
>  SP           | 140 +- 52.9        | 193 +- 11.5               | 233 +- 11.5          |
>
>
> - Secondly, we compared the causal action matrices $\mathcal{M}$ learned from different behavior policies (Appendix D.7, Table 8). We note that although using data collected by MEP yields better performance than data collected by Llama-8B, this is not MEP is inherently a “better” policy, but because MEP is more deterministic, making it easier to model the causal action matrix given the same number of steps. Importantly, even with data from Llama-8B, incorporating causal knowledge still provides measurable performance gains over baselines without causal guidance.
>
> **Together, these points demonstrate that the benefits of CausalPlan are not solely dependent on the specific behaviour policy used for data collection.**
>
>
> # Q10: $a_t$ is “unobservable”
>
> We clarify that the phrasing in the original manuscript was misleading. $a_t$ is observable during training (since we use trajectories to fit the model) but unobservable during validation/inference, when the agent must predict the next action without access to ground-truth $a_t$. We have reworded this in the revised manuscript (line 227-229) to make the distinction clear.

---

> ### Author Response · Authors · 2025-11-19
> **Reply Reviewer RXj8 (4/4)**
>
> # Q8: Binary Feature Encoding
>
> **Binary feature encoding is a common assumption in prior work on causality-integrated reinforcement learning, where the contribution is on improving planning capability rather than disentangling the environment (e.g., [2,3,4,8,9])**. It provides a tractable abstraction of the environment that captures the presence or absence of relevant factors, which is sufficient for learning the causal relationships relevant to next-action selection in our Overcooked-AI setup.
>
> **While this representation does discard some fine-grained information such as exact spatial distances, timing, or detailed interaction history, the key dependencies for planning and coordination—such as whether an object is available or whether a partner is carrying an item—are preserved in the binary features**. This trade-off between expressiveness and tractability  allows the SCA model to learn effective causal patterns without being overwhelmed by unnecessary detail.
>
> [8] Seitzer, M., Schölkopf, B., \& Martius, G. (2021). Causal influence detection for improving efficiency in reinforcement learning. Advances in Neural Information Processing Systems, 34, 22905-22918.
>
> [9] Chen, W., Zhang, J., Zhu, H., Xu, B., Hao, Z., Zhang, K., ... \& Cai, R. (2025). Causal-aware Large Language Models: Enhancing Decision-Making Through Learning, Adapting and Acting. Proceedings of the International Joint Conference on Artificial Intelligence (IJCAI).
>
> # Q9: Causal Graph G
>
> We have added an explicit visualization and discussion of the learned causal graph to Appendix E.1 (of the revised manuscript). To evaluate whether our method recovers meaningful structure, we validate it against the known recipe (domain knowledge) and physical constraints defined by the Overcooked-AI rules. We use the Cramped Room (CR) layout for illustration because it contains the smallest set of actions/features, making the learned graph easy to interpret.
>
> 1. Recipe Graph (domain knowledge)
>
> - To construct the recipe graph, we begin from the soft causal action matrix (learned under the MEP behavior policy; Appendix Fig. 10). We then remove any bidirectional edges by keeping only the direction with the higher learned weight, yielding a directed acyclic graph (penalize length-2 cycles). The resulting structure correctly identifies the canonical recipe sequence required by the environment:
> pickup_onion → put_onion_in_pot → pickup_dish → fill_dish_with_soup → deliver_soup, followed by a return to pickup_onion. These edges (highlighted in red in the figure) match the ground-truth procedural dependencies of the cooking task.
>
> - The other edges (black) also show that the matrix $M$ do not directly correspond to the MEP policy but are plausible indirect dependencies arising from the structure of the recipe. For example, while a trained model like MEP does not execute “pickup_onion → pickup_dish” consecutively, such a dependency is reasonable because a dish is typically picked up only after onions have already been placed in the pot, and the model captures this broader sequencing.
>
>
> 2. Physical Constraint
>
> - To illustrate physical constraint, we look at local causality. We specifically look at action “pickup_onion”. Applying a hard intervention with a threshold of 0.7 (keeping only edges with scores above the threshold) identifies the relevant state features that causally influence this action: empty_hand1 and empty_hand2. This is consistent with task logic, as the agent can only pick up an onion when one of its hands is free. While some additional edges are present, they do not contradict task logic and may reflect broader dependencies; importantly, their inclusion is associated with improved task performance. This demonstrates that even at the level of individual actions, the learned causal action matrix $M$ captures meaningful dependencies between state features and next actions, supporting the interpretability and correctness of the inferred graph. We politely refer the reviewer to Appendix C.8 for an illustration of a physical constraint failure mode that is successfully handled by the learned causal graph.

---

### Official Review · Reviewer_Tiqn · 2025-11-04

**Soundness:** 3
**Presentation:** 3
**Contribution:** 2
**Rating:** 6
**Confidence:** 2

**Summary:**

This paper proposes CausalPlan, a framework designed to improve LLM-based multi-agent collaboration by incorporating explicit causal reasoning into the planning process. The method introduces a Structural Causal Action (SCA) model that learns a causal graph from offline trajectories, modeling dependencies between state factors, prior actions, and next action choices. During inference, the causal graph is used to reweight sampled candidate actions from the LLM, promoting causally consistent planning and filtering out invalid or incoherent actions.
Experiments are conducted on the Overcooked benchmark showing consistent improvements.

**Strengths:**

1. The paper clearly identifies a real pain point in the LLM-planning space  LLM agents often rely on correlations and fail under causal inconsistencies  and proposes a targeted solution. The motivation aligns well with current challenges in multi-agent LLM systems.
2. The method section is well-structured and easy to follow. The paper clearly explains the proposed causal model and the way it integrates with LLM action sampling. The theoretical argument that, under standard identifiability assumptions, the causal graph and functional relationships can be uniquely recovered adds credibility and supports why the approach should work.
3. Implementation details are provided in good depth, including model architecture and prompting strategies.

**Weaknesses:**

1. It does not compare against the most recent LLM-agent + causal reasoning  methods. For example, CausalMACE[1] and Causal-aware LLMs[2].
2. All evaluations are done in the Overcooked kitchen environment. While this benchmark is standard, it is still a fairly constrained action/state space in a stylized cooperative setting. It would be helpful to see results in a more diverse or general multi-agent domain (e.g., social games, robotics simulators). Otherwise, it's unclear how easily the method generalizes to richer or more realistic scenarios.
3. The method depends on manual factorization of state/action features, and lower-level actions are ignored. This raises concerns about domain specificity and manual engineering effort. In complex environments, designing semantic factors may be non-trivial, and it’s unclear how the method scales without strong prior knowledge.


[1]https://aclanthology.org/2025.findings-emnlp.777/

[2]https://www.ijcai.org/proceedings/2025/0478.pdf

**Questions:**

1. The paper claims not to rely on the LLM’s causal reasoning ability, yet the pipeline still depends heavily on the LLM for analysis and candidate-action generation via a two-prompt design and knowledge library. Could the authors clarify whether the method truly disentangles causal reasoning from linguistic reasoning? To what extent could the observed gains stem from improved prompting workflow rather than causal modeling itself?
2. Each decision step requires: extracting factorized features and querying the causal matrix, is the runtime overhead significant?
3. Does the training buffer come from the same task distribution as evaluation? Are trajectories from them fully disjoint from test episodes? Could the causal structure overfit to the demonstration policy rather than reflect true task dynamics?

---

> ### Author Response · Authors · 2025-11-19
> **Reply Reviewer Tiqn (1/4)**
>
> We sincerely thank the reviewer for their careful reading of our work and for constructive feedback. We appreciate the thoughtful points regarding both the strengths and limitations of our approach. We hope that the following clarifications and additions address the concerns of the reviewer and strengthen the contribution of our work.
>
> # W1: Comparison with Causal-Aware LLMs and CausalMACE.
>
> - We thank the reviewer for referring to these papers. In fact, these are very strong works demonstrating the integration of causality and LLM-based decision making.
> - **We also want to clarify that we did compare with Causal-Aware LLMs [1], as referenced in Lines 322–323 and results in Appendix D of the original manuscript and line 346-348 of the revised manuscript**. This method was not originally designed for the Overcooked-AI environment and does not natively support multi-agent zero-shot planning. To ensure a fair comparison, we adapted our evaluation to their setup. Our adapted comparison confirms that CausalPlan achieves consistent improvements even under their experimental configuration. **We also clarify the main difference between our method and Causal-aware LLMs in line 513-516 of the revised version, stating that the this method "directly providing the causal graph as part of the LLM prompt", whereas we modify planning through aligning the decoding step of LLM with a causal score.**
>
> - Regarding CausalMACE [2], the code for this paper is not yet publicly available (see https://github.com/qccq315/CausalMACE). Therefore, we missed including it in the initial submission. **We have now added the following discussion and comparison to CausalMACE in the Related Work section (line 520-523) of the revised manuscript**. To provide further details: "CausalMACE prompts an LLM to explicitly construct a causal graph from task descriptions and game rules, relying on the model’s linguistic reasoning to infer causal dependencies. In contrast, CausalPlan does not ask the LLM to build or infer a causal graph. Instead, it learns a data-driven causal influence model over state, action to next-action transitions directly from interaction trajectories. This avoids dependence on LLM-generated causal structures, enabling CausalPlan to operate reliably even in multi-agent settings where rule descriptions are incomplete, ambiguous, or unavailable."
>
>
> [1] Chen, W., Zhang, J., Zhu, H., Xu, B., Hao, Z., Zhang, K., ... & Cai, R. (2025). Causal-aware Large Language Models: Enhancing Decision-Making Through Learning, Adapting and Acting. arXiv preprint arXiv:2505.24710.
>
> [2] Chai, Q., Zheng, Z., Ren, J., Ye, D., Lin, Z., & Wang, H. (2025). Causalmace: Causality empowered multi-agents in minecraft cooperative tasks. arXiv preprint arXiv:2508.18797.

---

> ### Author Response · Authors · 2025-11-19
> **Reply Reviewer Tiqn (2/4)**
>
> # W2: Justification of Benchmark Choice (Overcooked and Crafter).
>
> - We thank the reviewer for this thoughtful comment. **Although our empirical evaluation is primarily conducted on Overcooked-AI, we do not view this as a limitation for several reasons:**
>
> 1. First, Overcooked-AI is one of the few benchmarks explicitly designed to evaluate generalizable multi-agent collaboration. Prior works—including both traditional RL approaches (e.g., COLE [3], MEP [4]) and recent LLM-based methods (e.g., ProAgent [5])—also exclusively rely on this environment.
>
> 2. Second, the structured layouts in Overcooked-AI allow us to isolate collaboration challenges. In the FC layout, which requires very tight synchronization, most LLMs perform poorly. This highlights the difficulty of precise timing and interleaving actions, which is hard to evaluate in other environments where tasks are less structured.
>
> 3. Finally, our goal is to assess the applicability of our method across LLMs of varying sizes similar to previous work on planning [6]: Evaluating multiple LLM backbones and partner policies is computationally expensive. For each of our 4 LLMs, we run 3 seeds × 5 partner policies × 5 layouts × 1 role (Player 1), yielding 75 experiments per model (300 total for Fig. 3) plus an additional 75 experiments for Llama3‑70B as Player 0 (Tab. 4). This surpasses prior work—e.g., ProAgent [5] reports results for a single LLM.
>
> - **In addition, we also evaluated our framework on the Crafter environment as mentioned in the previous section (see Appendix E) — a widely used benchmark for assessing causal reasoning and long-horizon credit assignment in grounded decision-making [1,7].**
>
> References:
>
> [3] Li, Y., Zhang, S., Sun, J., Du, Y., Wen, Y., Wang, X., & Pan, W. (2023, July). Cooperative open-ended learning framework for zero-shot coordination. In International Conference on Machine Learning (pp. 20470-20484). PMLR.
>
> [4] Zhao, R., Song, J., Yuan, Y., Hu, H., Gao, Y., Wu, Y., ... & Yang, W. (2023, June). Maximum entropy population-based training for zero-shot human-ai coordination. In Proceedings of the AAAI Conference on Artificial Intelligence (Vol. 37, No. 5, pp. 6145-6153).
>
> [5] Zhang, C., Yang, K., Hu, S., Wang, Z., Li, G., Sun, Y., ... & Yang, Y. (2024, March). Proagent: building proactive cooperative agents with large language models. In Proceedings of the AAAI Conference on Artificial Intelligence (Vol. 38, No. 16, pp. 17591-17599).
>
> [6] Qiao, S., Fang, R., Zhang, N., Zhu, Y., Chen, X., Deng, S., ... & Chen, H. (2024). Agent planning with world knowledge model. Advances in Neural Information Processing Systems, 37, 114843-114871.
>
> [7] Hu, X., Zhang, R., Tang, K., Guo, J., Yi, Q., Chen, R., ... & Chen, Y. (2022). Causality-driven hierarchical structure discovery for reinforcement learning. Advances in Neural Information Processing Systems, 35, 20064-20076.

---

> ### Author Response · Authors · 2025-11-19
> **Reply Reviewer Tiqn (3/4)**
>
> # W3: Manual factorization and one hot encoded.
>
> We agree that this is a valid observation. Our design currently relies on manual factorization and one‑hot binary encodings for both states and actions. **However, assuming a discretizable environment is a common assumption in prior works studying causal reinforcement learning and causal LLM planning, where the contribution is on improving planning capability rather than disentangling the environment [1,7,8,12]** (mentioned in line 193-194 of the original manuscript). **We have added the following discussion in the revised version (Appendix C.1.1 of the revised version)**:
> - "Moreover, the assumption of discrete variables and state-action factorization do not restrict the broader applicability of our approach. Many practical domains—such as recommendation systems and text-based planning agents—naturally produce structured, vector-based observations that can be discretized or directly mapped to symbolic variables. Our evaluation closely reflects real-world scenarios like Model Context Protocol (MCP) servers, where agents receive textual prompts and select appropriate API function calls. Likewise, text-based recommendation systems process natural language inputs about user preferences to generate textual suggestions. These examples share the symbolic, text-in/text-out framework of Overcooked-AI, underscoring the practical relevance and generalizability of our approach beyond simulated settings. Additionally, a number of methods have been developed to recover symbolic factors from high‑dimensional inputs for the purpose of causal discovery: for example, CausalVAE [9], DEAR [10], and more recently VLM-based approaches learn disentangled latent state‑action representations, while VACERL [11] demonstrates effective causal discovery directly in image‑based environments. The main limitation in moving to continuous space lies in the complexity of learning a mapping function that must be learned during causal modeling. "
>
> References:
>
> [8] Seitzer, M., Schölkopf, B., & Martius, G. (2021). Causal influence detection for improving efficiency in reinforcement learning. Advances in Neural Information Processing Systems, 34, 22905-22918.
>
> [9] Yang, M., Liu, F., Chen, Z., Shen, X., Hao, J., & Wang, J. (2021). Causalvae: Disentangled representation learning via neural structural causal models. In Proceedings of the IEEE/CVF conference on computer vision and pattern recognition (pp. 9593-9602).
>
> [10] Shen, X., Liu, F., Dong, H., Lian, Q., Chen, Z., & Zhang, T. (2020). Disentangled generative causal representation learning.
>
> [11] Nguyen, M. H., Le, H., & Venkatesh, S. (2024, August). Variable-Agnostic Causal Exploration for Reinforcement Learning. In Joint European Conference on Machine Learning and Knowledge Discovery in Databases (pp. 216-232). Cham: Springer Nature Switzerland.
>
> [12] Yu, S., & Lu, C. (2024). Adam: An embodied causal agent in open-world environments. arXiv preprint arXiv:2410.22194.
>
> # Q1:  Disentangles causal reasoning from linguistic reasoning
>
> - We thank the reviewer for raising this important point. **To clarify, in CausalPlan we do not prompt the LLM to perform causal reasoning or causal discovery—this is precisely what disentangles causal reasoning from linguistic reasoning.** This separation of causal and linguistic reasoning distinguishes our approach from causal-aware LLMs [1] and CausalMACE [2].  The LLM is used only for linguistic tasks: generating high-level action candidates and retrieving task-relevant knowledge from the library. **All causal reasoning—both causal modeling and plan scoring—is handled entirely by our external causal modeling pipeline, independent of the internal reasoning capabilities of the LLM.**
>
>
> - To demonstrate the separation between improvement by prompt and improvement by CausalPlan, we explicitly investigated the contributions of the LLM prompting versus the causal modeling components in CausalPlan through an ablation study in Table 2.
> In this study, we compared a single-prompt configuration in ProAgent [5] — where one prompt handles both observation analysis and planning — with our two-prompt configuration, which separates analysis from planning. This comparison helps to isolate whether performance gains arise from improved prompting or from embedded causal knowledge. As reported, **performance between the single-prompt and two-prompt configurations is nearly identical**, showing that linguistic reasoning alone in the analysis prompt is not sufficient to achieve the full benefits of CausalPlan. We further evaluated the contribution of the Causal Backup Plan module. Removing this module, CausalPlan still outperforms the two-prompt baseline by 27%, but falls short of the full framework by 7%, which includes the Causal-Aware Planning component. **This demonstrates that the observed gains primarily stem from the causal components rather than from prompting.**

---

> ### Author Response · Authors · 2025-11-19
> **Reply Reviewer Tiqn (4/4)**
>
> # Q2: Inference Time Overhead
>
> **Due to page limits, we did not include this in the main paper but it is available in Appendix C.10 of the original manuscript in Appendix D.10 of the revised**, which provides a detailed time efficiency analysis. Learning the causal graph is a one-time process that can be mitigated with early stopping and re-use for different backbones. Inference overhead varies with model size. For 400 timesteps:
>
> Gemma-7B and LLaMA-8B: Runtime increases from ~5 → ~15 minutes.
> Qwen-14B: ~16 minutes → ~41 minutes.
> LLaMA-70B: ~40 minutes → ~68 minutes.
> **Importantly, the ratio increase is lower for larger models (1.7 times compared to 3 times for smaller models), which is a positive sign—these models are already computationally intensive, so the additional overhead introduced by our method remains acceptable**. Meanwhile, smaller models have shorter inference times overall, so the absolute overhead they incur is also reasonable. These overhead is reasonable given roughly the 20% improvement in performance and the qualitative gains in safety and adherence to domain constraints. We agree that this trade-off should be highlighted in the conclusion.
>
> # Q3: Overfit to Demonstration Policy
>
> - To verify that CausalPlan does not overfit to a particular behavior policy, we provide two pieces of empirical evidence. First, in our main experiment, the training buffer is collected from the trajectories generated by an MEP agent paired with itself. During the evaluation, CausalPlan is applied across all LLM backbones paired with a variety of partners, including agents not seen during training. This ensures that the learned causal structure is generalized beyond the demonstration policy. **As reported in Table 5 (Appendix D.4 of the revised manuscript) for AI-AI evaluation, CausalPlan outperforms the baseline used for data collection (MEP) on 7 out of 10 testing tasks and consistently surpasses all other baseline agents. In the AI-Human evaluation (Figure 4), CausalPlan outperforms the MEP baseline on all 10 testing tasks.**
>
> - Second, in Appendix D.7 (Tab. 8), we present an ablation study using LLaMA-8B to collect data. **Even when using LLaMA-8B for data collection, incorporating causal knowledge still significantly improves performance compared to not using causal guidance at all (average score increases from 110.7 ± 12.8 to 125.3 ± 30.7 on CR task).**

---

### Official Review · Reviewer_gMwk · 2025-11-04

**Soundness:** 2
**Presentation:** 2
**Contribution:** 2
**Rating:** 2
**Confidence:** 4

**Summary:**

This paper identifies a key failure mode in LLM-based agents, particularly smaller models, which often generate causally invalid actions in multi-agent collaborative tasks. To address this, the authors propose **CausalPlan**, a two-phase framework. In Phase 1, "Causal Action Structure Learning," a Structural Causal Action (SCA) model is learned from a dataset of agent trajectories to capture the influence of previous actions ($a_{t-1}$) and current states ($s_t$) on the next action ($a_t$). This is stored in a Causal Action Matrix (M). In Phase 2, "Agent Planning with Causal Knowledge," this matrix M is used to guide the LLM's action selection. This is done via two modules: 1) "Causal-Aware Planning," which re-weights the LLM's output probabilities with the causal scores from M, and 2) "Causal Backup Plan," a fallback mechanism that greedily selects the highest-scoring causal action if the LLM fails to produce a valid one. Experiments on the Overcooked-AI benchmark and Crafter demonstrate that CausalPlan reduces invalid action.

**Strengths:**

1.	The proposed two-phase framework is intuitive and modular.
2.	The paper is easy to follow.
3.	The empirical results are extensive and show consistent performance gains across multiple LLM backbones (Gemma, Llama, Qwen) and evaluation settings (AI-AI collaboration and Human-AI collaboration), outperforming baselines on the Overcooked benchmark.

**Weaknesses:**

1. The paper's primary claim rests on "causality-driven planning". However, the SCA model learns a supervised mapping from $(s_t, a_{t-1})$ to $a_t$ based on data collected from a single behavior policy (MEP). It is highly questionable whether this process discovers true "causal" structure as defined by Pearl or simply learns the strong correlations and biases within that specific policy's data. The proof of identifiability (Proposition 1) relies on strong, standard assumptions (e.g., causal sufficiency, additive noise) that are difficult to justify in a complex, dynamic environment like Overcooked.

2. A major limitation, which is not adequately discussed, is that the Causal Action Matrix $M$ appears to be learned **per layout**. The heatmaps in Fig. 10 and 11 are specific to the "CR layout", and the offline training takes 3 hours per environment. This severely limits the method's scalability and flexibility, which is one of the primary advantages of using LLM-based agents. The authors provide no evidence or discussion on whether $M$ learned on one layout can generalize to another.

3. The central idea of learning an external model from trajectory data to score and refine LLM-generated plans is not novel. The paper's related work section is missing key work [1] on this specific problem.
-	ReAd [1] directly tackles the same problem of inefficient LLM grounding in multi-agent environments like Overcooked.
-	The proposed "Structural Causal Action (SCA) model" is conceptually very similar to the local advantage function used in [1]. Both frameworks learn a function from agent trajectory data (collected from a behavior policy $\pi_\beta$ here) to score the utility of the proposed plan. While this paper formulates the scorer as a causal model $P(a_t | s_t, a_{t-1})$, ReAd [1] formulates it as an RL-based advantage function, the high-level approach of using a learned, data-driven scorer to refine LLM plans is highly overlapping. The authors must discuss this and other related works to properly situate their contribution.

**Questions:**

1.	Could the authors please clarify the novelty of the SCA model compared to other data-driven refinement models, such as ReAd [1] ? A thorough comparison in the related work section is necessary.
2.	Can the authors provide more evidence that the SCA model is learning true causal relationships rather than just the strong policy-specific correlations from the MEP dataset? What happens if a sub-optimal or random policy is used to generate the dataset $B$?
3.	Does the Causal Action Matrix M learned for one layout (e.g., Cramped Room) have any utility when transferred to another layout (e.g., Asymmetric Advantages)? If not, doesn't this per-layout offline training requirement undermine the zero-shot generalization promise of using LLMs?

[1] Zhang, Y., Yang, S., Bai, C., Wu, F., Li, X., Wang, Z., & Li, X. (2024). Towards efficient llm grounding for embodied multi-agent collaboration. ACL 2025

---

> ### Author Response · Authors · 2025-11-19
> **Reply Reviewer gMwk (1/3)**
>
> # W1: True causal relationships versus policy-specific correlations
>
> - We acknowledge that the SCA model is trained on trajectories collected from a specific behavior policy (MEP) and agree that strictly speaking, our method does not recover the causal structure of the environment in the Pearl sense. To avoid any misunderstanding, we have updated the manuscript, specifically in the Introduction (lines 93–97), to clarify this distinction. We explicitly state that:
> **"Importantly, the SCA model characterizes causal influence at the policy-level within the agent’s decision process, rather than causal dynamics at the environment-level in the Pearl sense~\citep{pearl2009causality}. Its purpose is not to model the true causal mechanisms of the environment, but to extract a stable and interpretable dependency structure that can guide and refine the LLM’s action selection."**
>
> - **These dependencies, however, go beyond simple correlations because the SCA explicitly simulates interventions over state/action features**. Specifically, the optimization of $\delta$ and $\eta$ simulates do-interventions on individual state and action features (see Appendix C.1 of the revised manuscript) to determine whether perturbing a feature consistently changes the predicted next action. This interventional signal:
>
>      - breaks spurious correlations that do not affect downstream actions,
>
>     -  identifies features that reliably influence action choice under the behavior policy, and
>
>     - produces a sparse set of directed dependencies, which we store in the causal action matrix $\mathcal{M}$
>
> - These interventional dependencies extend beyond simple correlation: two features may be highly correlated in the dataset, but if intervening on one does not change the predicted action distribution, the SCA removes the corresponding edge. **Thus, while the SCA does not capture the causal dynamics at the environment-level, it does capture the reliable cause-effect relationships at the policy-level in how actions are generated—providing exactly the structure needed to improve LLM planning.**
>
> - Regarding the assumptions, we agree that the identifiability proof (Proposition 1) relies on standard but strong assumptions—such as causal sufficiency and additive noise—which may not hold perfectly in a complex, partially observed, and dynamic environment like Overcooked. Our identifiability result should only be viewed as a conceptual or theoretical grounding rather than a strict guarantee for real-world environments. The goal of the proof is to clarify what the SCA would recover under idealized conditions and to motivate our use of sparsity-inducing penalties and interventional perturbations during training.  Crucially, despite potential violations of these assumptions in practice, we observe that **the learned SCA remains stable across different agents** (see Appendix D.9 of the revised manuscript) and generalizes across layouts and partners (Table 1). For example, although the SCA was trained using MEP data, CausalPlan improves performance when cooperating with all other agents. This stability is not only conceptual but also **translates into improved downstream performance**: integrating the SCA into the LLM planner consistently yields better coordination, higher task success, and fewer invalid actions compared to baselines without causal guidance (Figure 3).

---

> ### Author Response · Authors · 2025-11-19
> **Reply Reviewer gMwk (2/3)**
>
> # Q2: Additional Empirical Evidence of Causal Relationships
> ## Evidence 1:
> To further demonstrate that the SCA recovers meaningful causal influence patterns—rather than overfitting to correlations from a single policy—we provide evidence showing cross-agent consistency of the learned causal matrix. In Appendix D.9 of the revised manuscript, we present the causal action matrix $\mathcal{M}$ learned from trajectories generated by the MEP expert policy. To test whether SCA is simply memorizing the statistics of this dataset, we trained a second SCA model using the trajectories of a fine-tuned LLaMA-8B agent, whose behavior is:
>
> - substantially less deterministic than MEP,
> - noticeably noisier and often sub-optimal, and
> - characterized by different temporal patterns and error modes.
>
> **Despite these substantial behavioral differences, the SCA trained in LLaMA-8B trajectories recovered structural patterns strikingly similar to the MEP-based SCA**. Although edge strengths are weaker—consistent with the higher stochasticity of LLM policy—directiveness, sparsity pattern, and key dependencies remain largely unchanged. This stability of the agent between agents indicates that **the SCA captures policy-invariant causal influence relationships, rather than memorizing correlations tied to a particular dataset.**
>
> ---
> ## Evidence 2:
> A second line of evidence comes from comparing the performance of CausalPlan with the underlying behavior policy used to learn $\mathcal{M}$. If the SCA merely encoded action-selection frequencies of the behavior policy (MEP), then CausalPlan’s performance would mirror MEP’s partner-dependent performance patterns. Instead, we observe the opposite. As shown in the table below, collected from experiments on the AA layout (where MEP underperforms while CausalPlan excels), the relative performance across partners differs markedly:
>
> - MEP performs best when paired with PBT, whereas both CausalPlan-Llama-8B and CausalPlan-Qwen-14B perform best with COLE.
> - The CausalPlan models perform better with SP than with FCP, even though MEP performs worse with SP than with FCP.
>
> | Partner | MEP | Llama-8B | Qwen-14B |
> |---------|---------|----------------|----------------|
> | COLE    | 153 ± 23.1 | 253 ± 11.5 | 253 ± 11.5 |
> | FCP     | 153 ± 46.1 | 186 ± 30.5 | 226 ± 23.1 |
> | PBT     | 187 ± 11.5 | 193 ± 23.1 | 240 ± 0 |
> | SP      | 140 ± 52.9 | 193 ± 11.5 | 233 ± 11.5 |
>
> **These differences clearly indicate that CausalPlan is not simply replicating MEP’s behavior, nor is its performance tied to the strengths or weaknesses of the policy used to collect training data.**

---

> ### Author Response · Authors · 2025-11-19
> **Reply Reviewer gMwk (3/3)**
>
> # Q1 & W3: Missing Related Work
>
> We thank the reviewer for pointing out the relation to ReAd [1]. While both approaches share the high-level idea of learning a scorer from offline trajectories to guide LLM planning, the techniques differ substantially:
>
> - **ReAd** learns an advantage critic and refines plans through advantage-weighted selection, providing a value-based signal of whether an action improves expected return.
> - **CausalPlan** learns a policy-level structural causal model (SCA): by intervening on inputs during training, it discovers which past actions and states causally influence the next action, yielding a sparse causal matrix that directly informs decoding.
>
> At inference, ReAd evaluates proposals through the critic, whereas CausalPlan enforces causal constraints by reweighting LLM log-probabilities and falling back to a causal greedy action when needed. **We have expanded the related work section in the revised manuscript (lines 488–496).**
>
>
> # Q3 \& W2 Per-layout training and generalization
>
> We acknowledge that the current implementation trains a separate SCA per layout. This does incur a per-layout offline cost (~3 hours for CR layout). However:
>
> - **This approach reflects a proof-of-concept demonstrating that causal structure can improve LLM-guided planning.**
> - **In our evaluation, we test SCA transfer across different partners for the same layout, showing robustness to partner behavior**. We also mention potential extensions to **layout-agnostic** in the revised manuscript line 538, highlighting this as a direction for future work.
> - While per-layout training may limit zero-shot generalization, **the improvement in action validity demonstrates that integrating causal knowledge is a promising avenue to enhance LLM planning**, which could be combined with LLM generalization methods.
>
> # Remarks
> We thank the reviewer for the thorough and insightful feedback. The points raised regarding the assumptions behind Proposition 1, the layout-specific nature of the Causal Action Matrix, and the relation to prior work such as ReAd [1] are very valuable. We have addressed these concerns in the revised manuscript by:
>
> - Clarifying that the SCA model captures **policy-level causal influence** rather than environment-level causal dynamics.
> - **We provide evidence of causation by showing that interventions on state and action features during training reliably identify dependencies that influence downstream actions, break spurious correlations, and improve coordination in multi-agent tasks. **
> - Expanding the related work section to highlight the similarities and differences between CausalPlan and data-driven refinement approaches such as ReAd [1].
>
> We hope these revisions address the reviewer’s concerns and clarify the novelty, evidence of causation, and practical utility of CausalPlan. We welcome any further feedback to improve the manuscript.

---

### Author Response · Authors · 2025-11-19

Dear Reviewers,

Thank you for taking the time to provide detailed feedback on our manuscript. **We have uploaded the revised version, with all modified content highlighted in red**. Overall, we have tried to address the questions and weaknesses raised in the previous version. In particular, we have clarified the causal interpretation of our method, distinguished it from missing related work, framed the appendix proof as a conceptual illustration with additional context on ridge regression and Bernoulli NLL, and provided empirical evidence regarding causation and policy correlation.

We hope these revisions make the contributions and findings clearer and more transparent, and we welcome further feedback to continue improving the manuscript.

---

### Meta-Review · Area_Chair_tagy · 2026-01-07

**Summary:**

This paper proposes CausalPlan, a framework to reduce invalid actions in LLM-based multi-agent collaboration. It learns a Structural Causal Action (SCA) model from offline trajectories to capture policy-level dependencies between states, previous actions, and next actions. This model is used to reweight the LLM's action probabilities during planning. Experiments in Overcooked-AI and Crafter demonstrate improvements in task success and reduced invalid actions for selected LLMs.

**Reviewer Concerns:**

The paper presents a system that delivers consistent empirical improvements on challenging benchmarks. However, the core "causal" framing has been walked back to "policy-level dependencies," and the method has scalability limitations (per-layout training). While the practical results are solid, the revised conceptual contribution—a form of structured policy distillation for action refinement—requies more clarification and feels somehow incremental given the existing literature.

**Reviewer Scores:**

The original ratings are relatively low for the papers in my stack. I thank the authors have made great efforts to improve this work. However, it seems that the average reviewer's ratings would borderline reject even if the ratings might change.

---

### Decision · Program_Chairs · 2026-01-26

Reject